# Machine Learning-Based Estimate of The Wind Speed Over Complex Terrain Using the LSTM Recurrent Neural Network

Cássia Maria Leme Beu[1,*] and Eduardo Landulfo[1,*]

[1]Instituto de Pesquisas Energéticas e Nucleares (IPEN), 2242 Prof. Lineu Prestes Av., São Paulo, Brazil
[*]These authors contributed equally to this work.

**Correspondence:** Cássia Beu (cassia.beu@gmail.com)

**Abstract.** Accurate estimate of the wind speed profile is crucial for a range of activities such as wind energy and aviation. The power law and the logarithmic-based profiles have been widely used as universal formulas to extrapolate the wind speed profile. However, these traditional methods have limitations in capturing the complexity of the wind flow, mainly over complex terrain. In recent years, the machine learning techniques have emerged as a promising tool for estimating the wind speed profiles. In this study, we used the Long Short-Term Memory (LSTM) Recurrent Neural Network and observational lidar datasets from three different sites over complex terrain to estimate the wind profile up to 230 m. Our results showed that the LSTM outperformed the Power Law as the distance from the surface increased. The coefficient of determination ($R^2$) was greater than 90% up to 100 m for input variables up to 40 m high only. However, the performance of the model improved when the 60 m wind speed was added to the input dataset. Furthermore, we found that the LSTM model trained on one site with 40 and 60 m observational data and applied to other sites also outperformed the Power Law. Our results show that the machine learning techniques, particularly LSTM, is a promising tool for accurately estimating the wind speed profiles over complex terrain, even for short observational campaigns.

## 1 Introduction

Machine learning techniques are increasingly being adopted as powerful tools in environmental sciences. We see many examples of this method applied for different purposes to forecast meteorological variables and their derivative products (Musyimi et al., 2022; Jiang et al., 2022; Mustakim et al., 2022; Jesemann et al., 2022). However, the use of the machine learning techniques is not restricted to the local or regional scales. Liu et al. (2022), for example, proposed a multi-level circulation pattern classification to identify large-scale weather or climate disaster events. The forecasting and monitoring disasters were also the subject of Soria-Ruiz et al. (2022). They got high performance by applying machine learning algorithms to remote sensing datasets to detect the recurrent floods over Gulf of Mexico coastline and the central and southeastern part of Mexico. Among the methods evaluated, Song and Wang (2020) concluded that the neural networks are superior to produce monthly wildfire predictions one year in advance, providing thus, a valuable information for long-range fire planning and management. Adding the Principal Component Analysis (PCA), Zhang et al. (2022) improved the accuracy for the visibility prediction at Sichuan (China). Among the six machine learning algorithms evaluated, they found out that the neural network performed best. Cheng

and Tsai (2022) proposed a hybrid methodology based on variable selection and autoregressive distributed lag to forecast the pollutant concentrations, which improved the results when compared to the full and without lag dataset. The Support Vector Regression (SVR), that is a supervised algorithm, performed better than the other four algorithms tested. Those are only few examples of innovative works adopting the machine learning techniques in the environmental sciences.

Wind forecasts underpin wind power prediction, which is essential to support wind energy production in the short-term. Although winds have been traditionally forecasted with Numerical Weather Prediction models, the use of Machine Learning has become more widespread, not only to correct the biases derived from the highly variable nature of the winds, but also as stand-alone prediction models. Wang et al. (2021) showed that their multi-layer cooperative combined forecasting system, which is based on a novel adaptive weighting scheme, overcame the limitations of the current single and combined forecasting methods and provided a more accurate and stable forecast. In their review paper, Bali et al. (2019) analyzed few studies produced during this century and concluded that the techniques for the wind speed forecast have limitations, as low efficiency and high computational cost. They proposed the use of Long-Short Term Memory (LSTM) to improve wind speed forecasting for power prediction. Tukur et al. (2022) analyzed works produced between 2010 and 2020 and concluded that ensemble and hybrid methods are reaching high accuracy, because they present more abilities to model complex functions than the linear models. They agreed with Bali et al. (2019) that the LSTM looks promising in forecasting the wind speed whilst recommending further investigation on the capabilities of hybrid model approaches. Dalton and Bekker (2022) showed the improvement when considering other meteorological variables into the modeling. Their results pointed to the vertical wind and divergence as important predictors to the wind speed. In this way, He et al. (2022) included the 2 m temperature and surface pressure to train their dual-attention mechanism multi-channel convolutional LSTM model with the ERA5 dataset to forecast the 10 m wind speed. Zhou et al. (2023) also used the ERA5 dataset to investigate the grid-to-site conversion models, considering altitude, land use and seasonality effects. The deep learning models outperformed the linear interpolation and the regression models to estimate the 10 m wind speed. The aforementioned works briefly exemplify that efforts have been applied to the wind speed forecast theme, however, the methods to estimate its vertical profile are still limited.

According to Pintor et al. (2022), extrapolating the wind speed to higher heights is still a challenge and from the two most widely used methods (the power law and the logarithmic-based profile) they found out that the first one is more accurate for a wide variety of landscapes. The Met Office (United Kingdom) developed the Virtual Met Mast (VMM) tool (Standen et al., 2016) to assess the wind profile, however, this technique requires high spatial resolution weather numerical prediction (Schwegmann et al., 2023). Only recently, machine learning techniques have been used to forecast the wind speed profile. Türkan et al. (2016) evaluated seven different machine learning methods to estimate the 30 m wind speed at Kutahya (Turkey) and concluded that the SVR produced the most realistic results than the other six. Al-Shaikhi et al. (2022) proposed the Particle Swarm Optimization (PSO) with the LSTM method and compared their results with other optimization algorithms for an experiment carried out at Dhahran (Saudi Arabia). Their model needs at least four different levels of observational data as input. Similarly, Nuha et al. (2022) proposed the Regularized Extreme Learning Machine (RELM) to extrapolate the wind speed to higher heights. With the same dataset of Dhahran, Mohandes and Rehman (2018) used the Restricted Boltzmann Machine (RBM) method and observations at four different heights as input. They showed that their method improved the wind speed

forecast. Bodini and Optis (2020a) and Bodini and Optis (2020b) found that random forests outperform standard wind extrapolation approaches, using a "round-robin" validation method. They highlighted the benefits of including observational data capturing the diurnal variability of the atmospheric boundary layer, namely the Obukhov length, Turbulence Kinetic Energy and time of the day, all of them measured at 4 m high. Vassallo et al. (2020) also improved their results including meteorological variables to the input dataset of their Artificial Neural Networks (ANN) model, advising to carefully select the input data

and emphasizing the importance of normalization. Even the VMM data are improved with machine learning methods (Schwegmann et al., 2023). Bodini and Optis (2020a) and Bodini and Optis (2020b) conducted their experiments over low complex terrain (Great Plains – US) and stressed the need of performing the same kind of analysis in more complex terrains. To the best of our knowledge, most studies on vertical wind speed extrapolation were conducted for low complexity orographies, except for Vassallo et al. (2020) who analyzed different types of terrain complexity, and Standen et al. (2016) and Schwegmann et al.

(2023) that conducted their studies through the VMM tool.

## 2  Data and methods

### 2.1  The LSTM Recurrent Neural Network

Recurrent Neural Networks (RNNs) are a type of artificial neural networks where the output of one time step is used as an input in the subsequent time step and then build a memory of time series events. The RNNs are specifically designed to work,

learn and predict sequential data (Medsker and Jain, 1999). The Long Short-Term Memory (LSTM) is a type of RNN that is considered a state-of-the-art tool for processing sequential and temporal data nowadays. The main advantage of the LSTM over the others RNNs is that the presence of internal memory allows maintaining long-term dependencies, avoiding the vanishing or exploding gradient problems (Smagulova and James, 2019). This was done by introducing a forget gate into the standard recurrent sigma cell of the RNNs. The forget gate can decide what information will be discarded (Yu et al., 2019) and makes

the LSTM system a robust model that compensates for the imperfections in the input data (Sherstinsky, 2020). The LSTM cells are mathematically expressed by:

$$f_t = \sigma(W_{fh}h_{t-1} + W_{fx}x_t + b_f) \tag{1}$$

$$i_t = \sigma(W_{ih}h_{t-1} + W_{ix}x_t + b_i) \tag{2}$$

$$\tilde{c}_t = \tanh(W_{\tilde{c}h}h_{t-1} + W_{\tilde{c}x}x_t + b_{\tilde{c}}) \tag{3}$$

$$c_t = f_t.c_{t-1} + i_t.\tilde{c}_t \tag{4}$$

$$o_t = \sigma(W_{oh}h_{t-1} + W_{ox}x_t + b_o) \tag{5}$$

$$h_t = \sigma_t \tanh(c_t) \tag{6}$$

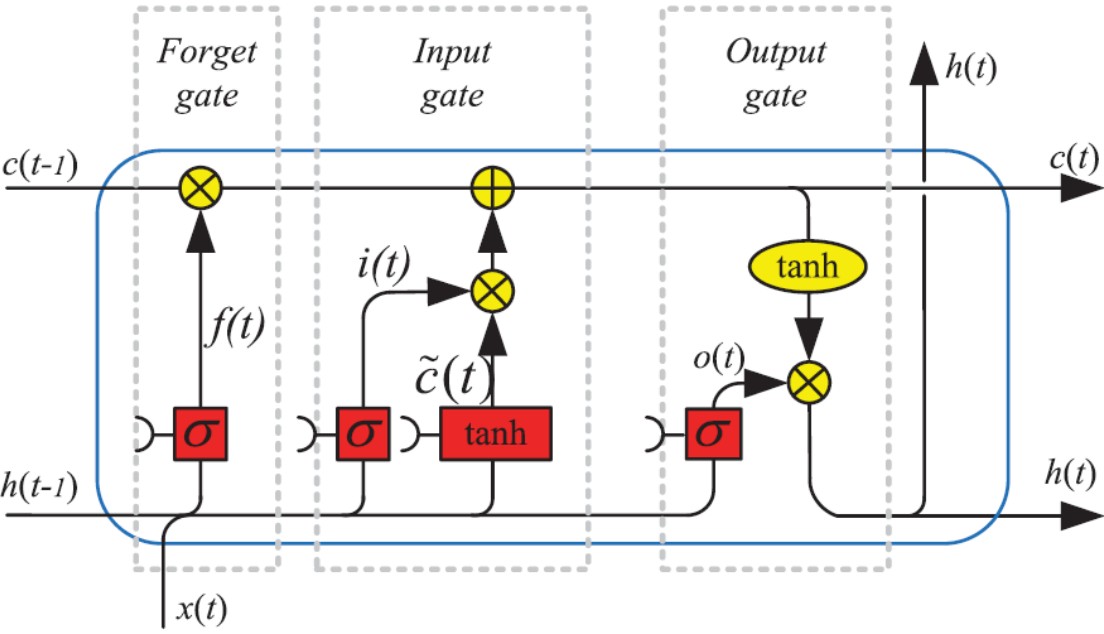

**Figure 1.** LSTM schematical diagram (Yu et al., 2019)

where $x_t$ and $h_t$ are the inputs and the recurrent information at time $t$; $c_t$ is the cell state of the LSTM; $f_t$, $i_t$ and $o_t$ are the forget, input and output gates; $W_f$, $W_i$, $W_{\tilde{c}}$ and $W_o$ are the weights; $b$ is the bias, the operator '.' is the pointwise multiplication of two vectors. Fig. 1 illustrates the LSTM compounds and architecture.

We run the LSTM using the Keras library (version 2.9) from Python (version 3.8.16) through Colab (the Google Research's platform). The missing data were interpolated using the interpolate Pandas function through a linear method. Afterwards, the data were normalized through the StandardScaler function from the Sklearn library (Pedregosa et al., 2011). The StandardScaler normalizes by removing the mean and scaling to the standard deviation:

$$z = (x - u)/s \tag{7}$$

where: $x$ is observed data, $u$ is the mean, $s$ is the standard deviation and $z$ is the normalized data.

We identified the optimal hyperparameters by using the Kerastuner (O'Malley et al., 2019) with the Hyperband algorithm. Table A1 exhibits the tuned hyperparameters for each experiment. We maintained the default configuration of Keras for the others LSTM arguments (Keras, 2023). See Table A2.

## 2.2 Doppler lidar

We employed the Windcube v2 Doppler lidar, from Leosphere, during the field campaigns at three different sites. For the Windcube v2 technical specifications, see Beu and Landulfo (2022). The information of the field campaigns are listed in Table 1.

**Table 1.** Information of the field campaigns

| SITE | ALTITUDE(m) | COORDINATES | OBSERVATIONAL PERIOD |
| --- | --- | --- | --- |
| 1 | 721 | -23.6;-46.7 | 18/Sept/2015 to 10/Mar/2016 |
|  |  |  | 11/Oct/2016 to 31/Dec/2016 |
| 2 | 4 | -23.9;-46.7 | 11/Mar/2016 to 25/Aug/2016 |
| 3 | 590 | -23.4;-47.6 | 26/Jul/2017 to 06/Aug/2018 |

The lidar was set up for 12 levels, as follows: 40, 60, 80, 100, 120, 140, 160, 180, 200, 230, 260 and 290 m and to retrieve information every 10 minutes. The Windcube v2 system automatically discards data that the Carrier-to-Noise (CNR) ratio is under -23 dB and we removed data that presented availability less than 80% over 10 minutes. See in Table A3 that the data availability is over 99% for all the three sites up to 160 m high. Above 160 m, the availability decreases to 98% at Site 2 and 94% at Site 3 at 230 m high.

We considered the observed data at 40 m to estimate the wind speed at higher heights (from 60 up to 230 m). Beyond the 10 min mean wind speed (v40), we also considered the wind direction (dir40), the hour, and the standard deviation of the horizontal ($\sigma$u + $\sigma$v) and vertical ($\sigma$w) wind speed to forecast the wind speed at higher heights. With the wind speed standard deviation, we estimated the Turbulence Kinetic Energy (TKE), which is the sum of the wind speed variances (Stull, 1988) and is expressed by:

$$TKE = \frac{1}{2}(\sigma_u^2 + \sigma_v^2 + \sigma_w^2) \tag{8}$$

As already found out, including cyclical variables improves the wind speed forecast (Bodini and Optis, 2020a, b; Baquero et al., 2022). The diurnal cycle is a strong feature of the sites under research and we will discuss this further. Since surface observations are not available, the 40 m TKE could indirectly transmit information related to temperature and stability, improving the modelling with respect to diurnal variability. This step is referred to **Experiment 1**. Afterwards, we also added the 60 m wind speed as input to forecast the heights above and it is referred to **Experiment 2**. Following the advice of Bodini and Optis (2020a) and Bodini and Optis (2020b), we conducted two more experiments (**Experiment 3** and **Experiment 4**) which consisted in swapping a trained model for another environment and to evaluate its performance. In this way, the trained model

for the Site 1 was applied to the Sites 2 and 3. As well, the trained model for the Site 2 was applied to Sites 1 and 3 and the trained model for the Site 3 was used for the Sites 1 and 2. In Table A4, we summarize the input variables of each experiment.

## 2.3 The Power Law

According to Pintor et al. (2022), the Power Law (PL) is the simplest and generally the most effective way to extrapolate the wind speed. The PL is given by:

$$V = V_r (\frac{z}{z_r})^\alpha \tag{9}$$

where, $V$ and $V_r$ are the wind speed at height $z$ and at reference height $z_r$, respectively. $\alpha$ is the wind shear coefficient. The authors state that: $\alpha < 0.1$, corresponds to unstable conditions; $0.1 < \alpha < 0.2$ is typical of neutral profile and $\alpha > 0.2$ describe a stable atmosphere.

## 2.4 Evaluation

For evaluating the model performances, we chose verification metrics like those used in Zhou et al. (2022) and Baquero et al. (2022), because those metrics have been largely applied to wind forecast through machine learning methods. For further information on these metrics, see Zhou et al. (2022) and Baquero et al. (2022).

· **Coefficient of determination** ($R^2$): The $R^2$ tells us how much the model differs from the original data and it is related to the correlation coefficient.

$$R^2 = 1 - \frac{\Sigma_{i=1}^{N}(y_i - \hat{y_i})^2}{\Sigma_{i=1}^{N}(y_i - \overline{y})^2} \tag{10}$$

· **Mean Squared Error (MSE)**:

$$MSE = \frac{1}{N}\Sigma_{i=1}^{N}(y_i - \hat{y_i})^2 \tag{11}$$

· **Root Mean Squared Error (RMSE)**:

$$RMSE = \sqrt{\frac{1}{N}\Sigma_{i=1}^{N}(y_i - \hat{y_i})^2} \tag{12}$$

· **Mean Absolute Error (MAE)**:

$$MAE = \frac{1}{N}\Sigma_{i=1}^{N}|y_i - \hat{y_i}| \tag{13}$$

· **Mean Absolute Percentage Error (MAPE)**:

$$MAPE = \frac{100\%}{N}\Sigma_{i=1}^{N}\frac{y_i - \hat{y_i}}{max(\epsilon, |y_i|)} \tag{14}$$

where $y_i$, $\overline{y}$ and $\hat{y}_i$ are: the actual value, the mean of the observed data and the predicted value. $N$ is the total number of data points and $\epsilon$ is an arbitrarily small but strictly positive number to avoid undefined results when $y_i$ is zero.

Lastly, we applied the bootstrapping technique (Efron and Tibshirani, 1994) to estimate the error bars for $R^2$. For this purpose, we used the bootstrap function from Scipy Library (Virtanen et al., 2020), with a confidence level of 0.95 and number of resamples equal to 100 times the data points.

## 2.5 Observational campaigns

The observational campaigns took place over a three year period (Table 1) on the southeastern portion of Brazil (Fig. 2). All
155 three observational sites are within 140 km from the coast, and clearly marked on the map. Despite the proximity between sites (see the description of Fig. 2), the types of terrain are completely different, namely the height and surface roughness (Table 1). The Site 1 is inside the Metropolitan Region of São Paulo, which is characterized by a densely mixed urban matrix.

Site 2 is a coastal municipality called Cubatão. Beyond the industrial zone, Cubatão is surrounded by natural parks of the Atlantic Rain Forest (Morellato and Haddad, 2000), residential areas and a high mountain range, called Serra do Mar, on
its north boundary. At this point, Serra do Mar rises sharply, up to more than 700 m high, across 5 km wide and acts as an important barrier to the atmospheric circulation. Vieira and Gramani (2015) provide a technical description of the Cubatão and Serra do Mar features.

Site 3, the Iperó municipality, is more than 130 km away from the coast, as shown in Fig. 2. It is inside a predominantly rural area and about 10 km away from the urban zone of the Sorocaba municipality. Another important characteristic of this Site is
165 the Araçoiaba Hill to the southeast, rising up more than 300 m high up to 900 m altitude. The Araçoiaba Hill is inside a Federal Conservation Unit called Ipanema National Forest.

## 3 Results

The surface strongly affects the atmospheric circulation within the Planetary Boundary Layer (PBL). Thus, we plotted the wind rose for the first observational level (40 m) as an attempt to identify similarities and differences among the three sites. In
this study, the wind rose shows the direction where the wind blows from (as typically used in meteorology). The circulation patterns are similar between Sites 1 and 3 (Fig. 3 and Fig. 5). Both of them present a diurnal cycle of winds turning $360^o$. We see this diurnal cycle in Fig. A1, which illustrates a 30 days wind direction temporal series. Most of the time, the wind turns along the day, except for short periods identified by the red circles, when the winds remain mainly from south-southeast and are related with postfrontal events. The sea breeze (southeast wind) is one of the main reasons for the pattern of Fig. 3 at Site 1
(Ribeiro et al., 2018). According to Ribeiro et al. (2018), there are two main conditions that inhibit the sea breeze reaching the São Paulo Metropolitan Region (SPMR): the prefrontal circulation and the cloudiness. The cloudiness decreases the thermal contrast between the sea and the land and the prefrontal circulation is opposed to the sea breeze. Thus, excluding those two conditions, the sea breeze advances over the SPMR often along the year and justifies the wind rose pattern (Fig. 3). Even at 40 m above the surface, the winds are weak and rarely reach 8 m/s. However, the Low-Level Jet (LLJ) is a typical feature of the

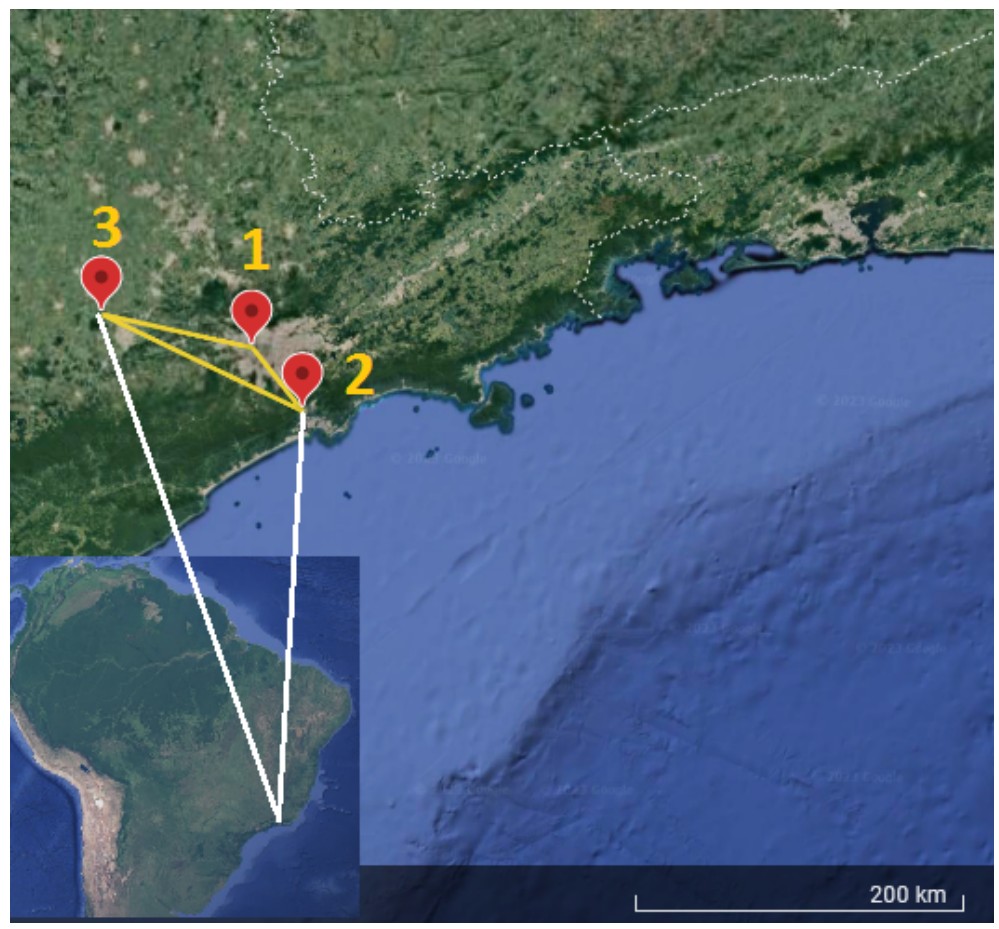

**Figure 2.** Sites of the observational campaigns. The distance (yellow line) is 47 km between Sites 1 and 2; 131 km between Sites 2 and 3; 90 km between Sites 1 and 3. Distance estimated by the Google Earth tool (© Google Earth)

SPMR (Sánchez et al., 2022) and the power and logarithmic law fail in extrapolating the wind speed profile in LLJ environment when compared to machine learning methods (Bodini and Optis, 2020a, b).

At Site 2 (Fig. 4), for this observational period, north and northeast winds were disproportionately more frequent than the other directions. However, it is also possible to identify a diurnal cycle, as observed in Fig A2. Except for the postfrontal events, identified by the red circles, the wind direction is variable along the day. Klockow and Targa (1998) illustrated a conceptual model (their Figure 2) and explained in a simplified way the local atmospheric circulation, where the sea and the land breezes play an important role. This frequent wind direction reversal, due to the sea-land contrast and the orography reported by Klockow and Targa (1998) may prejudice the model performance. Compared to the Site 1, the wind speed is weaker. Vieira-Filho et al. (2015) also observed a similar pattern of Fig. A2 (rotating $360^o$ throughout the day) for the surface winds and emphasized the influences of the orography and the ocean on the local circulation. They detected around 20% of calms (wind speed < 1 m/s), occurring preferably at nighttime and mean wind speed around 2.4 m/s.

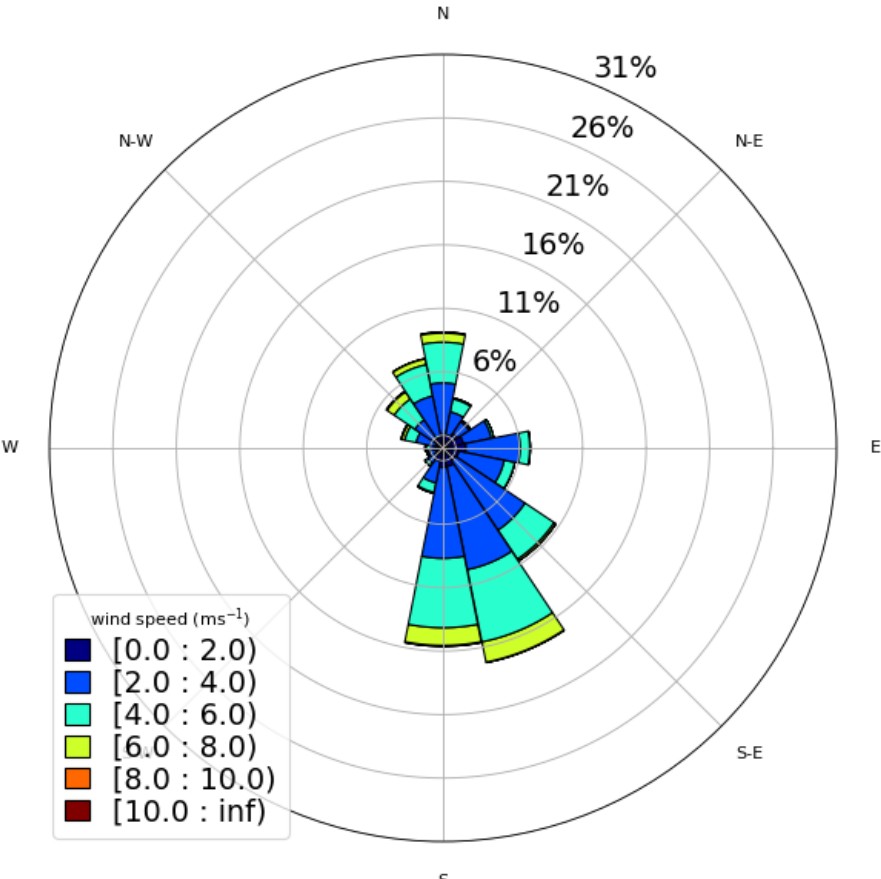

**Figure 3.** Observed wind at 40 m - Site 1 (normalized wind rose). The wind speed is indicated by legend (m s$^{-1}$)

The diurnal cycle at Site 3 (Fig. A3) is mainly related to the mountain-valley circulation since the valley (Tietê river valley) becomes deeper to northwest. Thus, the local circulation generally turns $360^o$ throughout the day, resulting in the wind rose shown in Fig. 5. See that the seasonal pattern in Fig. A4 is comparable to that in Fig. 5. The circulation is also influenced by the frontal passages and the postfrontal condition generates stronger south and southeast winds than the prefrontal that generates weaker north and northwest winds. The LLJs are a recurrent feature observed at this Site (de Oliveira et al., 1995) and can form very near the surface (Beu and Landulfo, 2022). Winds are slightly stronger than the other two sites, but rarely reach 10 m/s (Fig. 5).

We carried out more than 60 experiments, testing different machine learning models with multiple configurations, namely: Random Forest Trees (Breiman, 2001) that was applied by Bodini and Optis (2020a) and Bodini and Optis (2020b); SVR (Smola and Schölkopf, 2004); its two different implementations - nuSVR and LinearSVR (Pedregosa et al., 2011); Multi-layer

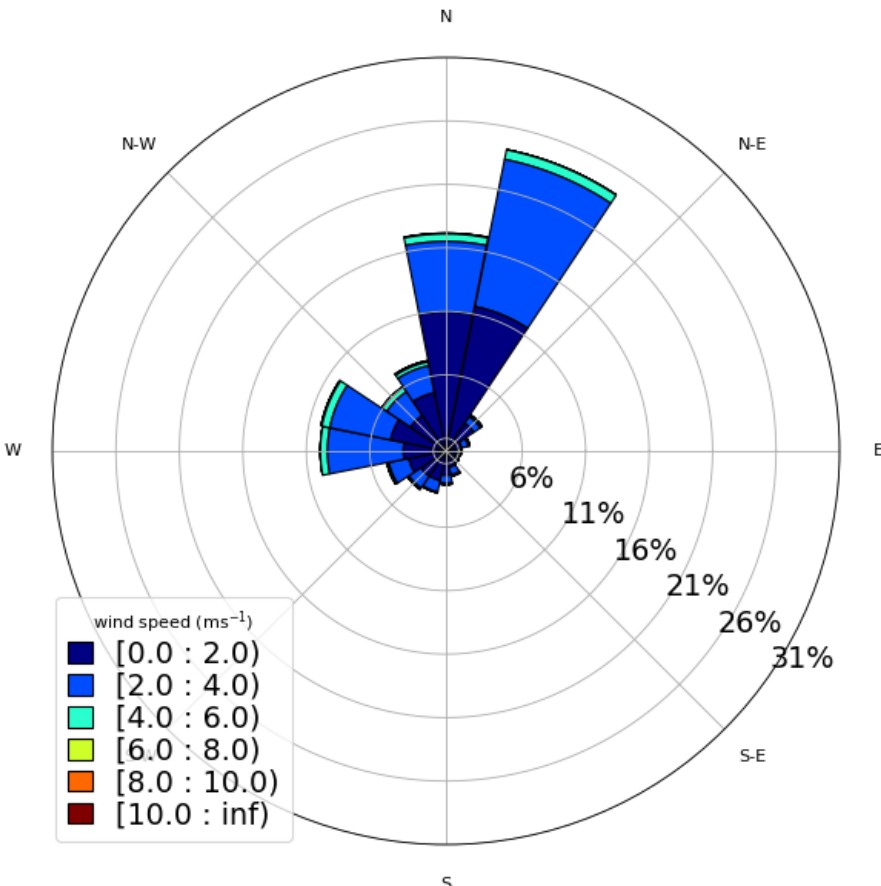

**Figure 4.** Observed wind at 40 m - Site 2 (normalized wind rose). The wind speed is indicated by legend (m s$^{-1}$)

Perceptron (Almeida, 1997); Complete Ensemble Empirical Mode Decomposition with Adaptive Noise - CEEMDAN (Torres et al., 2011). At their work, Türkan et al. (2016) used the SVR, the Multi-layer Perceptron and and the Random Forest Trees algorithms. CEEMDAN (Torres et al., 2011) has already been applied for wind speed forecasts (Wang et al., 2021). The LSTM RNN outperformed the SVR, nuSVR, LinearSVR, Random Forest Trees, Multi-layer Perceptron and CEEMDAN. Results also improved when 10 min mean data were used as input instead of 30 min mean or 1 hour mean. Here we only present results for the best performing model LSTM RNN (Bali et al., 2019; Al-Shaikhi et al., 2022).

### 3.1 Experiment 1

Data from Site 3 was first used to train the model; starting with wind speeds at 40 m to predict speeds at higher heights. The entire dataset contains more than 50 thousand data points for each variable. As we were working at Colab, for each new test it

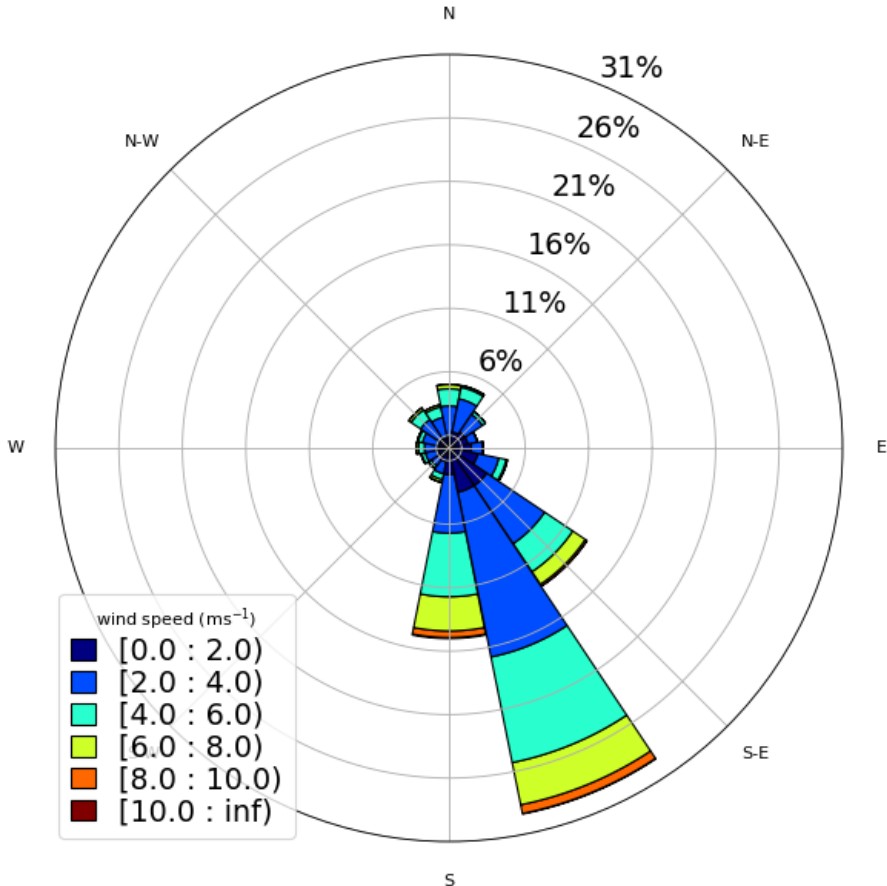

**Figure 5.** Observed wind at 40 m - Site 3 (normalized wind rose). The wind speed is indicated by legend (m s$^{-1}$)

was necessary to upload the dataset. Inputing the whole dataset for training and testing the model consumes much processing time. Considering we were working at Google Colab, for each new test it was necessary to upload the dataset again. Despite this, running machine learning on Colab is advantageous, in the sense that many libraries are easily accessible and don't require installation at the local machine. The Google Colab also ease the team work, since the code can be safely shared with the group members. Surprisingly, we found that model improvement plateaued without using all of the data points of record. During this

phase we conducted tests changing the dataset size and hyperparameters and evaluated the improvement through the metrics (Eq. 10 to 14).

For the Site 3, we found that the ideal dataset size was 8 thousand data points, taking 90% for the training. As the time series is comprised of 10 min temporal averages, that corresponds to roughly two months of observational data. We followed testing the inclusion of other variables, as: wind direction, hour and TKE (Eq. 8), because those data give information about the diurnal

cycle and improved the model. Tables 2, 3 and 4 and Fig. 6, 7 and 8 present the results reached by the LSTM model and the Power Law (PL), according to Eq. 9 and $\alpha = 0.25$, as we found that this value provides the best correlation for our datasets. See Table A1 for the dataset sizes and hyperparameters. For all three sites, the $R^2$ is similar for estimates with the PL and the LSTM at the first level (60 m) however, as the distance from the surface increases, the LSTM estimates outperform the PL. That behaviour was also observed by Liu et al. (2023). This happens because the PL has a universal nature and cannot simulate

features as the LLJ (Bodini and Optis, 2020a).

**Table 2.** Site 1 - Assessment of the wind speed estimated by the PL and the LSTM model (**Experiment 1**) up to 230 m

| Height (m) | Observed mean wind speed (m s$^{-1}$) | Model | $R^2$ | MSE (m$^2$s$^{-2}$) | RMSE (m s$^{-1}$) | MAE (m s$^{-1}$) | MAPE (%) |
|---|---|---|---|---|---|---|---|
| **60** | 3.61 | PL | 0.98 | 0.06 | 0.25 | 0.20 | 6.47 |
| | | LSTM | 0.98 | 0.05 | 0.22 | 0.17 | 6.63 |
| **80** | 3.85 | PL | 0.94 | 0.18 | 0.42 | 0.33 | 10.74 |
| | | LSTM | 0.95 | 0.13 | 0.36 | 0.28 | 10.20 |
| **100** | 4.01 | PL | 0.90 | 0.30 | 0.54 | 0.43 | 13.06 |
| | | LSTM | 0.93 | 0.21 | 0.46 | 0.35 | 11.43 |
| **120** | 4.17 | PL | 0.86 | 0.45 | 0.67 | 0.53 | 15.06 |
| | | LSTM | 0.91 | 0.31 | 0.56 | 0.43 | 13.15 |
| **140** | 4.29 | PL | 0.82 | 0.62 | 0.77 | 0.62 | 17.08 |
| | | LSTM | 0.88 | 0.41 | 0.64 | 0.50 | 14.64 |
| **160** | 4.41 | PL | 0.76 | 0.82 | 0.91 | 0.72 | 19.32 |
| | | LSTM | 0.84 | 0.53 | 0.73 | 0.57 | 16.23 |
| **180** | 4.52 | PL | 0.69 | 1.07 | 1.03 | 0.82 | 21.71 |
| | | LSTM | 0.81 | 0.66 | 0.81 | 0.63 | 18.17 |
| **200** | 4.64 | PL | 0.60 | 1.39 | 1.18 | 0.93 | 24.31 |
| | | LSTM | 0.76 | 0.83 | 0.91 | 0.70 | 19.80 |
| **230** | 4.75 | PL | 0.48 | 1.86 | 1.37 | 1.08 | 28.19 |
| | | LSTM | 0.70 | 1.05 | 1.02 | 0.78 | 22.96 |

     For the Site 1, we reached the best result with a temporal series with 10 thousand data points. This is approximately a 70 days observational campaign. When only 40 m variables are used as predictors, we obtain $R^2 > 90\%$ up to 120 m (Table 2). The MSE and MAE also confirm the superiority of the LSTM model over the PL. Even the MAPE is greater for the PL estimates than for the LSTM estimates.

**Table 3.** Site 2 - Assessment of the wind speed estimated by the PL and the LSTM model (**Experiment 1**) up to 230 m

| Height (m) | Observed mean wind speed (m s$^{-1}$) | Model | $R^2$ | MSE (m$^2$s$^{-2}$) | RMSE (m s$^{-1}$) | MAE (m s$^{-1}$) | MAPE (%) |
|---|---|---|---|---|---|---|---|
| **60** | 2.30 | PL | 0.96 | 0.09 | 0.30 | 0.21 | 12.8 |
| | | LSTM | 0.97 | 0.08 | 0.28 | 0.20 | 12.6 |
| **80** | 2.50 | PL | 0.89 | 0.36 | 0.60 | 0.41 | 22.6 |
| | | LSTM | 0.93 | 0.22 | 0.47 | 0.35 | 20.7 |
| **100** | 2.69 | PL | 0.81 | 0.88 | 0.94 | 0.60 | 30.3 |
| | | LSTM | 0.91 | 0.43 | 0.66 | 0.46 | 26.1 |
| **120** | 2.91 | PL | 0.72 | 1.83 | 1.35 | 0.81 | 36.7 |
| | | LSTM | 0.88 | 0.77 | 0.88 | 0.59 | 28.5 |
| **140** | 3.15 | PL | 0.65 | 3.08 | 1.75 | 1.01 | 41.7 |
| | | LSTM | 0.89 | 0.97 | 0.98 | 0.66 | 29.4 |
| **160** | 3.35 | PL | 0.61 | 4.23 | 2.06 | 1.17 | 46.2 |
| | | LSTM | 0.88 | 1.27 | 1.13 | 0.74 | 31.7 |
| **180** | 3.52 | PL | 0.58 | 5.11 | 2.26 | 1.31 | 49.8 |
| | | LSTM | 0.89 | 1.39 | 1.18 | 0.79 | 33.8 |
| **200** | 3.70 | PL | 0.53 | 6.61 | 2.57 | 1.47 | 52.6 |
| | | LSTM | 0.86 | 2.02 | 1.42 | 0.88 | 36.4 |
| **230** | 3.86 | PL | 0.50 | 7.79 | 2.79 | 1.61 | 55.3 |
| | | LSTM | 0.80 | 3.04 | 1.74 | 1.05 | 38.1 |

Comparing the Site 1 and the Site 3, we see through Fig. 6 and Fig. 8 that the PL performance decreases faster at Site 3 than at Site 1. At 160 m, the $R^2 = 62\%$ at the Site 3 and $R^2 = 76\%$ at the Site 1. For the Site 2, the PL performance also exhibits fast decreasing with the height (Fig. 7), similarly to the Site 3. Summarizing, we see from Fig. 6, 7 and 8 that the PL works better close to the surface. Looking at the scatter plot for the Site 1, we see that the PL performance compares to the LSTM (Fig. 9) at 60 m, but at 230 m, we see stronger winds underestimated by the PL (the red circle on Fig. 10).

The Site 2, which has weaker winds (see Table 3, column 2), presents better performance for the LSTM forecast from 140 m upwards than the other two sites. As shown by Fig. 7, $R^2$ remains almost constant above 140 m, while for the PL, the $R^2$ decreases faster than the Site 1 curve. The PL underestimates winds stronger than 8 m/s as illustrated by the scatter plot (Fig. 11) and are associated with abrupt changes as indicated by the temporal series (Fig. 12). The causes of that strengthening of the wind profile are unknown and remain as suggestion for a future investigation. The LSTM also underestimates the stronger

**Table 4.** Site 3 - Assessment of the wind speed estimated by the PL and the LSTM model (**Experiment 1**) up to 230 m

| Height (m) | Observed mean wind speed (m s$^{-1}$) | Model | $R^2$ | MSE (m$^2$s$^{-2}$) | RMSE (m s$^{-1}$) | MAE (m s$^{-1}$) | MAPE (%) |
|---|---|---|---|---|---|---|---|
| **60** | 4.59 | PL | 0.96 | 0.22 | 0.47 | 0.36 | 10.0 |
| | | LSTM | 0.98 | 0.09 | 0.30 | 0.22 | 7.1 |
| **80** | 5.03 | PL | 0.89 | 0.64 | 0.80 | 0.62 | 14.9 |
| | | LSTM | 0.96 | 0.21 | 0.46 | 0.34 | 10.0 |
| **100** | 5.37 | PL | 0.83 | 1.10 | 1.05 | 0.83 | 18.1 |
| | | LSTM | 0.95 | 0.34 | 0.58 | 0.44 | 11.8 |
| **120** | 5.68 | PL | 0.77 | 1.64 | 1.28 | 1.01 | 20.8 |
| | | LSTM | 0.92 | 0.60 | 0.77 | 0.58 | 13.6 |
| **140** | 5.93 | PL | 0.71 | 2.21 | 1.49 | 1.16 | 22.8 |
| | | LSTM | 0.89 | 0.81 | 0.90 | 0.67 | 14.5 |
| **160** | 6.16 | PL | 0.62 | 2.94 | 1.71 | 1.33 | 25.1 |
| | | LSTM | 0.86 | 1.11 | 1.05 | 0.78 | 15.4 |
| **180** | 6.36 | PL | 0.54 | 3.67 | 1.92 | 1.48 | 27.0 |
| | | LSTM | 0.83 | 1.39 | 1.18 | 0.90 | 17.5 |
| **200** | 6.52 | PL | 0.47 | 4.33 | 2.08 | 1.61 | 28.7 |
| | | LSTM | 0.80 | 1.66 | 1.29 | 0.97 | 18.5 |
| **230** | 6.75 | PL | 0.37 | 5.35 | 2.31 | 1.79 | 30.9 |
| | | LSTM | 0.78 | 1.83 | 1.35 | 1.04 | 19.3 |

winds (mainly the winds that exceed 12 m/s), as we see from the scatter plot, but it captures the pattern better than the PL (Fig. 12).

The metrics show a similar behavior between Site 1 and Site 3. Despite the complex topography, perhaps the better performance of the LSTM model for the Site 2 for the levels above 140 m is related to the absence of the LLJ. To the best of our knowledge, LLJs so close to the surface have not been reported there yet; on the contrary, they are a common feature of the Sites 1 and 3 (Sánchez et al., 2022; de Oliveira et al., 1995; Beu and Landulfo, 2022).

## 3.2 Experiment 2

Some studies (e.g., Vassallo et al., 2020; Mohandes and Rehman, 2018) already showed that adding input variables from different heights below the extrapolation height improves the machine learning performances. Thus, we added the 60 m wind speed

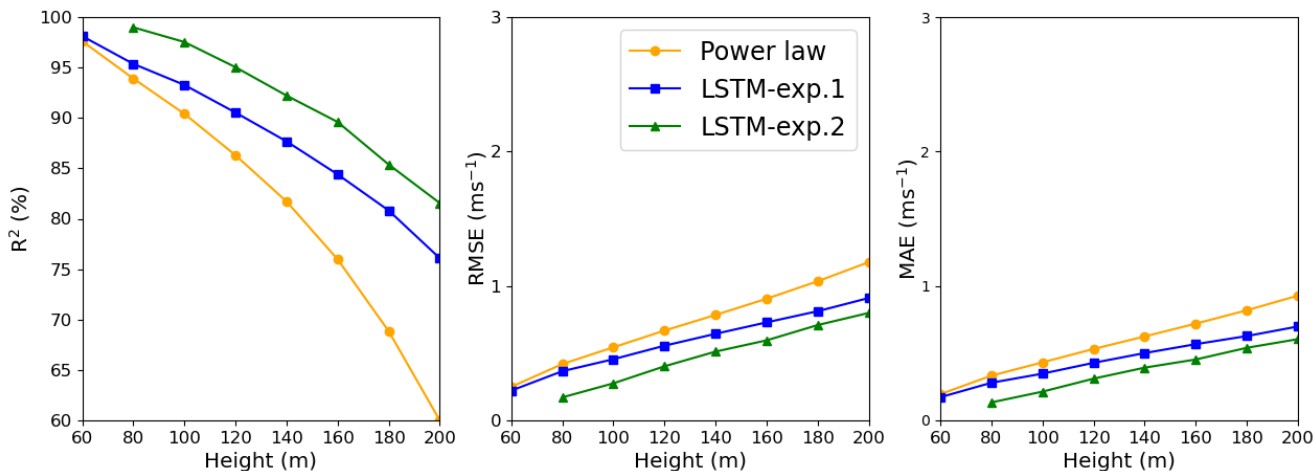

**Figure 6.** LSTM and Power Law $R^2$, $RMSE$ and $MAE$ estimates: Site 1. Exp.1 and exp.2 stand for experiment 1 and experiment 2, respectively

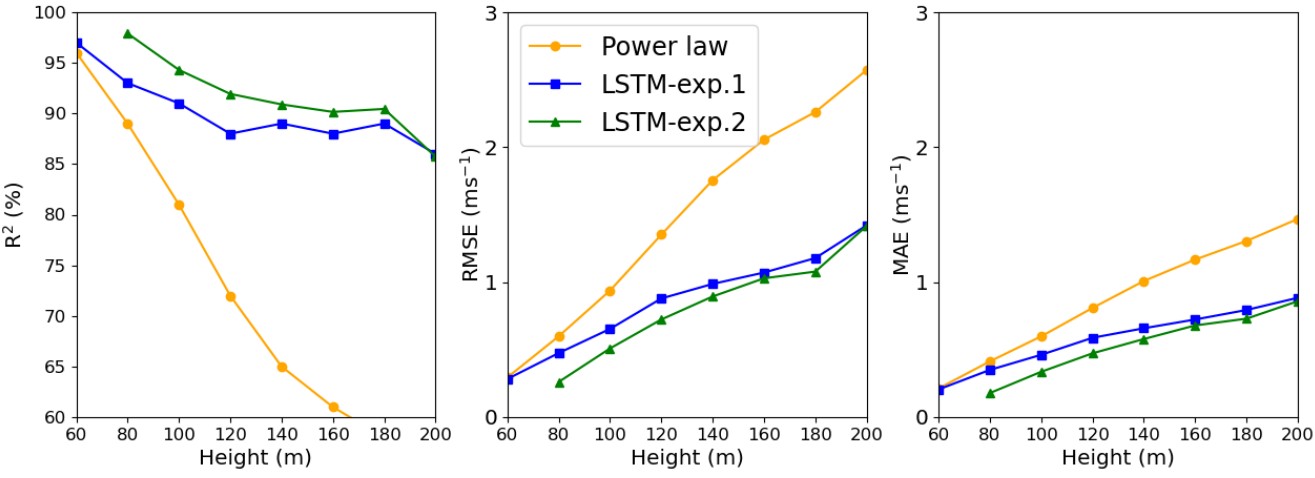

**Figure 7.** LSTM and Power Law $R^2$, $RMSE$ and $MAE$ estimates: Site 2. Exp.1 and exp.2 stand for experiment 1 and experiment 2, respectively

observations to the input dataset of the experiment 1 to estimate the above heights. Adding the 60 m wind speed observations to the input dataset improved the results, as we see in Fig. 6, 7 and 8 (green line). For the Site 1 we see an increasing along the entire $R^2$ curve, reaching 99% at 80 m, while the MAE decreased by 50%. At 200 m, the $R^2$ increased by 6% and the MAE reduced more than 8%. The improvement was more pronounced at the lower heights for the Site 2 (compare the blue and green lines in Fig. 7). The $R^2$ increased to 98% against the 93% from Experiment 1 at 80 m and the MAPE was reduced by 70%, but for the higher levels, the improvement gradually decreases, as we see from Fig. 7.

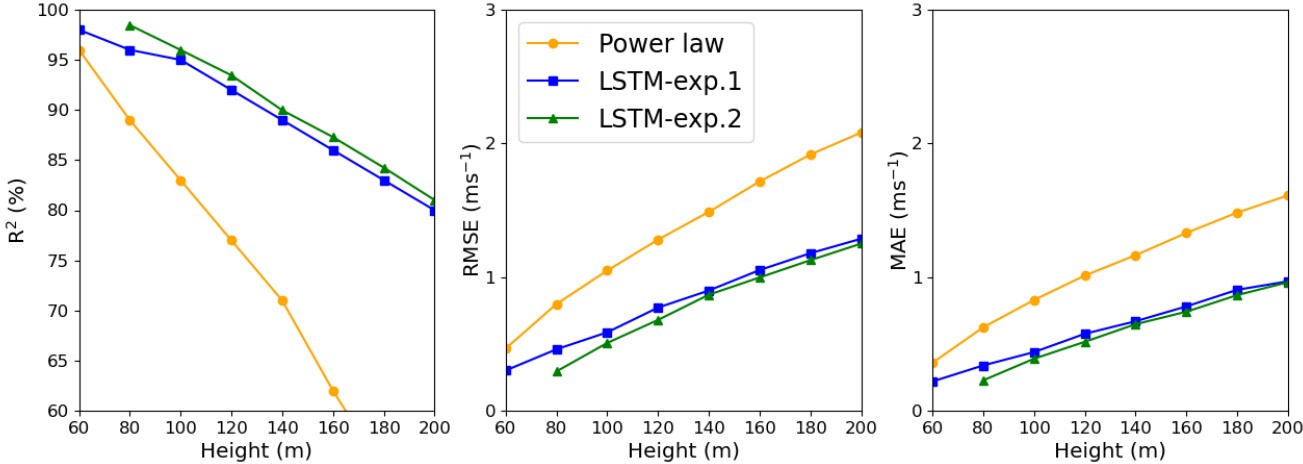

**Figure 8.** LSTM and Power Law $R^2$, $RMSE$ and $MAE$ estimates: Site 3. Exp.1 and exp.2 stand for experiment 1 and experiment 2, respectively

For the Site 3, the Experiment 2 also outperformed the Experiment 1 and the improvement is constant with the height, just slightly better at 80 m as we see from the greater distance between the green and blue lines (Fig. 8). The $R^2$ increased 2.5% at 80 m and only 1.5% at 200 m. Performing the bootstrapping method (Fig. A5, A6, A7), is evident that the variability is higher for the PL estimate for the sites 2 and 3, mainly. For the Site 1, despite the LSTM-exp 2 exhibited similar error bars, the $R^2$, is higher.

We also found out that the performance is kept if we change the sample size for the tests. For the tests, we evaluated the $R^2$ for three different samples beyond those from Table A1. Tests were done for 2000, 4000 and 7000 data points. For the PL estimate, the $\alpha$ was computed taking the 40 and 60 m wind speed. The results for the 80 m and 180 m forecasts are shown in Fig. A8. We see that the $R^2$ are comparable for the LSTM and PL estimates and remain almost constant for the 80 m forecast for the tests conducted with 2000, 4000 and 7000 data points at Site 1. For the 180 m forecast, the LSTM performance slightly

increases when the sample dataset increases from 2000 to 7000. For the Site 3 case, the 80 m forecast presented only slight variation and LSTM and PL performances are also comparable. The Site 2 forecasts exhibited a slight improvement when the test dataset increased from 2000 to 7000 data points at 80 m. In this case, the PL performance was worse than the LSTM. The 180 m forecast for the Site 2 is not shown because the original dataset reported an atypical pattern, with the 180 m wind speed weaker than the 40 m wind speed. Because of that atypical pattern, even the PL estimate failed. The PL estimate at 180 m for

the Sites 1 and 2 were also worse than the LSTM forecast and are not shown because are out of the figure scale.

### 3.3 Experiment 3

Bodini and Optis (2020b) advised about the importance of applying the machine learning models to different sites of that where they were trained. Following their advice, we applied each trained model to the other two sites (Fig. 13 - Fig. 15).

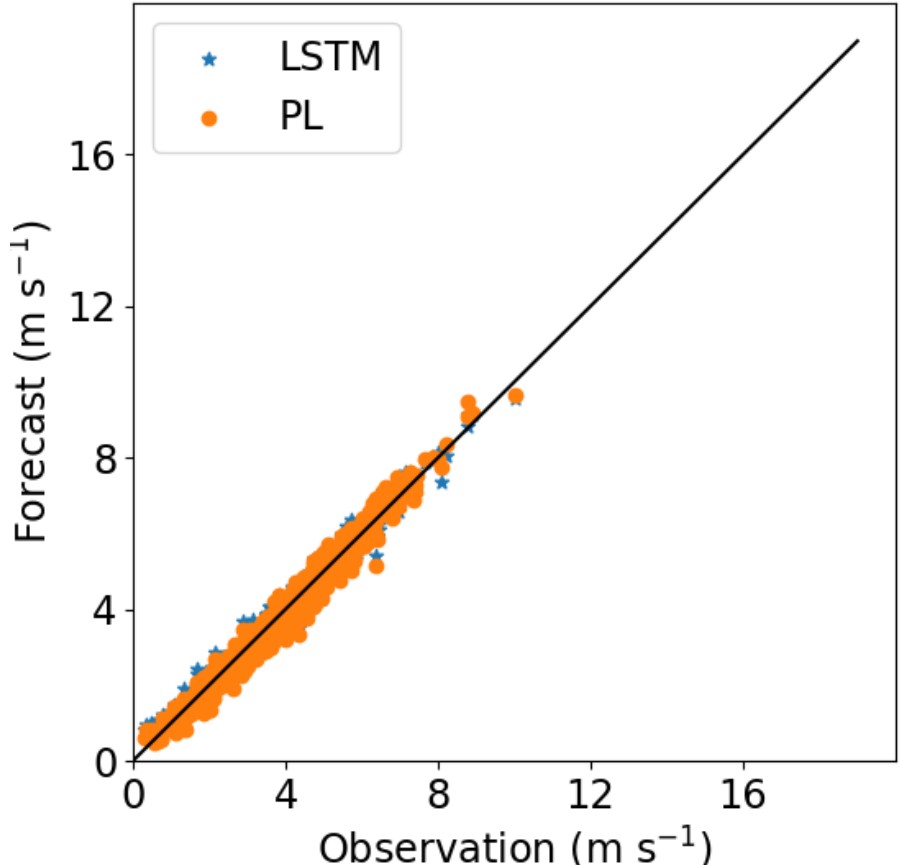

**Figure 9.** LSTM and PL forecasts (Site 3) for 60 m – results from **Experiment 1**

For the Site 1 (Fig. 13) we see that the Site 3 model (blue line) performed better than the Site 2 model (green line), but its performance was worse than the original model (S1, that was trained and validated at the Site 1). It is also clear from this figure that the performance quickly decreases with the height. The behaviour is the same for the Site 3 (Fig. 15), where the model trained for the Site 2 presented the worst result. The tests of the models trained at Site 1 and Site 3 for the Site 2 presented poor performance as indicated by the fast $R^2$ reduction with the height (Fig. 14).

Figures 16 - 18 show the correlation between observed and forecasted wind speed for 80 m, 100 and 140 m for the forecast of the Site 1 with the model trained at Site 3.

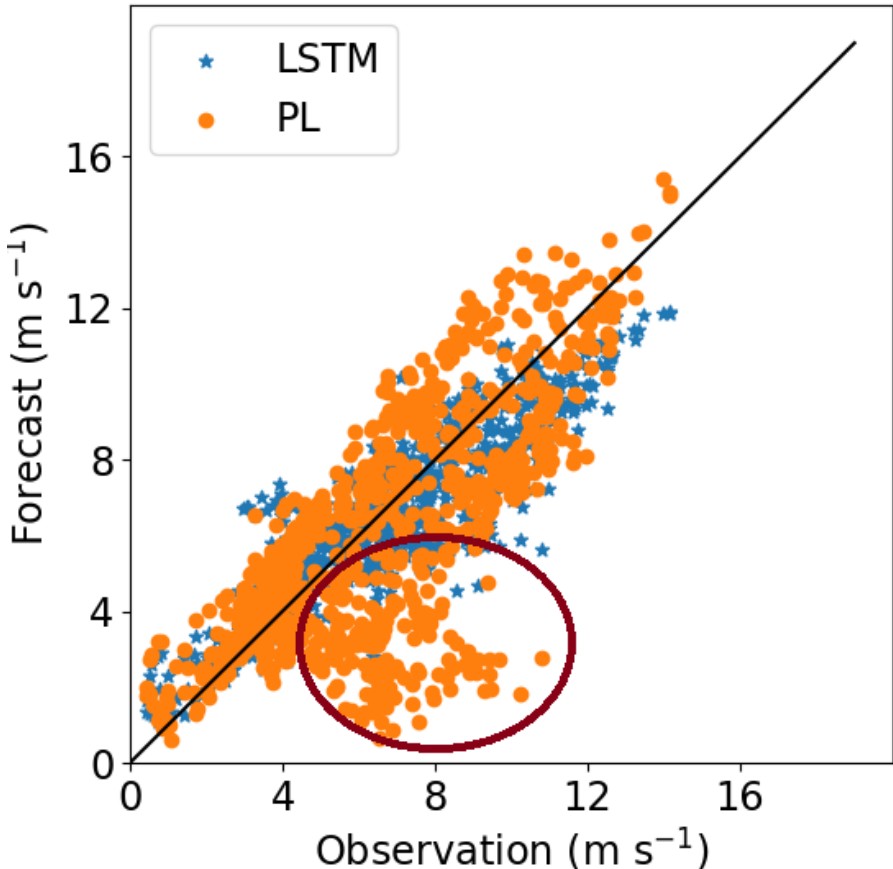

**Figure 10.** LSTM and PL forecasts (Site 3) for 230 m – results from **Experiment 1**

### 3.4 Experiment 4

For this step, we took the best result from the previous experiment (**Experiment 3**) and added the 60 m wind speed to the input dataset. That means, for the Site 1 case, took the model trained at Site 3.

The forecast for the Site 1 highly improves when the 60 m wind speed is included on the input dataset for training the model at Site 3, as we see in Fig. 19 and outperforms the PL forecast. The $R^2$ increased by 7% if compared with the LSTM forecast with only the 40 m observations (**Experiment 3**) for the 80 m height. The $R^2$ reached 90.6% and 84.9% at 120 and 140 m, respectively. This result is almost as good as **Experiment 1**. Figures 22 - 24 illustrate the improvement (compared to Fig. 16 - 18) when the 60 m wind speed observation was added to the training phase.

We also observe a strong improvement for the Site 3 (Fig. 21) compared to the PL estimate. At 80 m, the $R^2$ increased by 9%

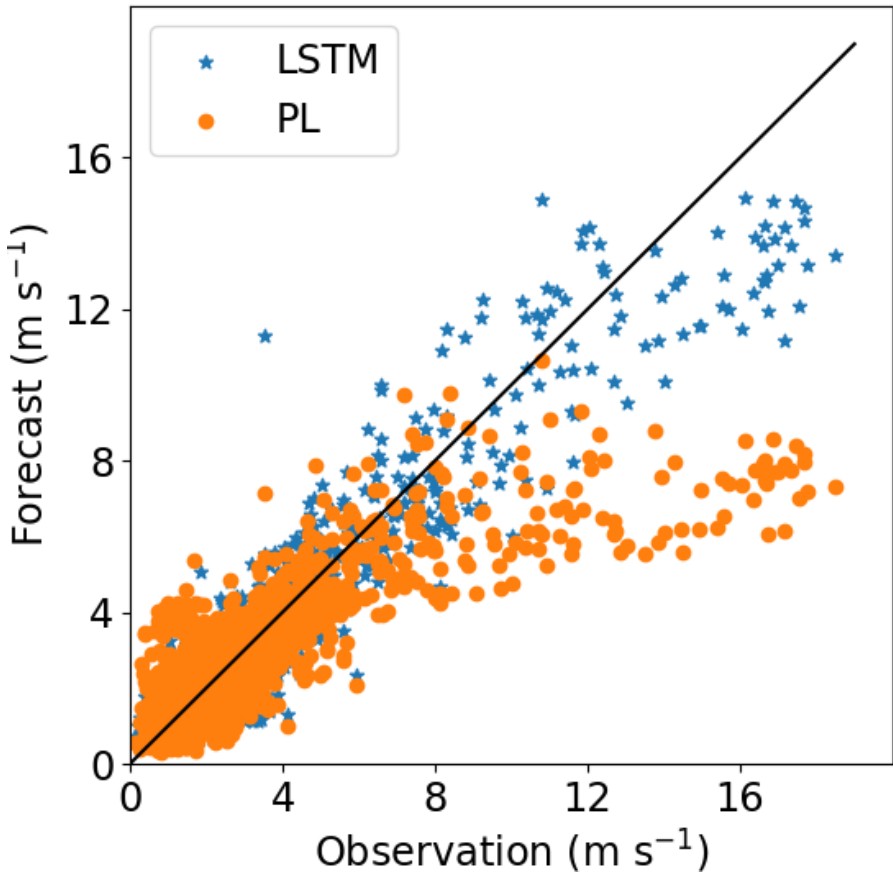

**Figure 11.** Site 2: 160 m wind speed forecast (**Experiment 1**)

compared to the PL estimate, while at 140 m, we observed an increase of 16%.

For the Site 2 we used the model trained at Site 1, since that one performed best, as indicated by Fig. 14. In this case, we see improvement up to 120 m (Fig. 20), but it was lesser than the other two cases. It is obvious that adding more observational levels to the input dataset would improve the results, however, it is not clear if this method should be applied if the surfaces are too different as the Site 2 in relation to Site 1 and Site 3. We recommend more tests for the complex terrain scenarios.

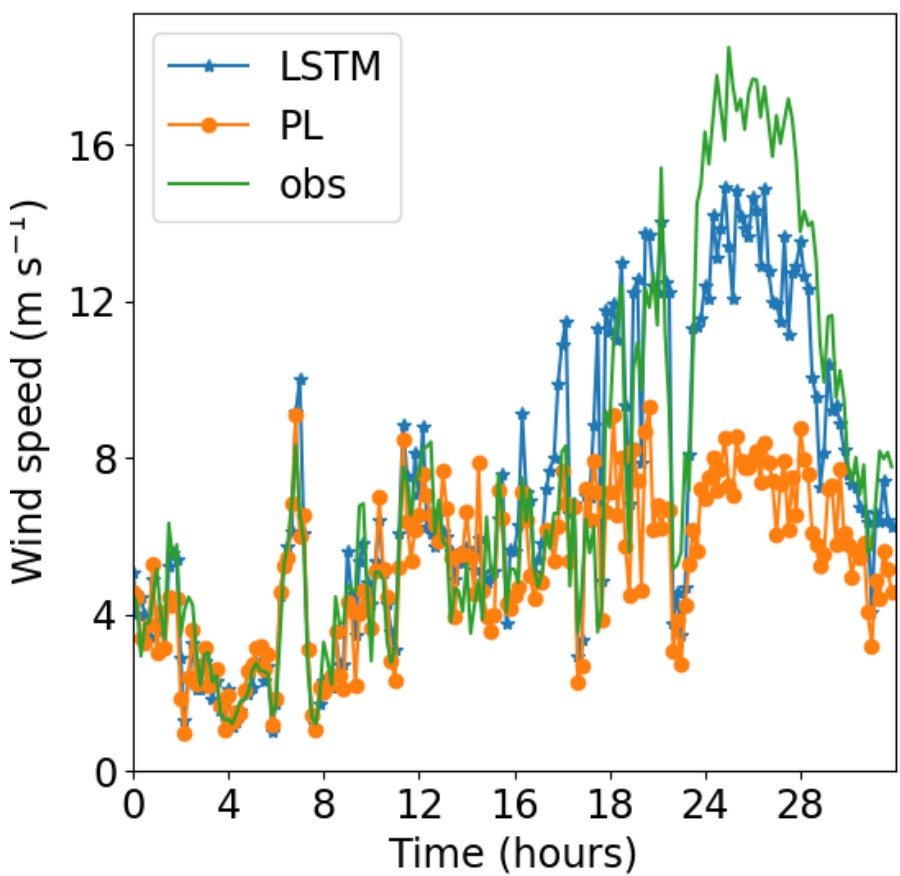

**Figure 12.** Site 2: 160 m wind speed temporal series (**Experiment 1**)

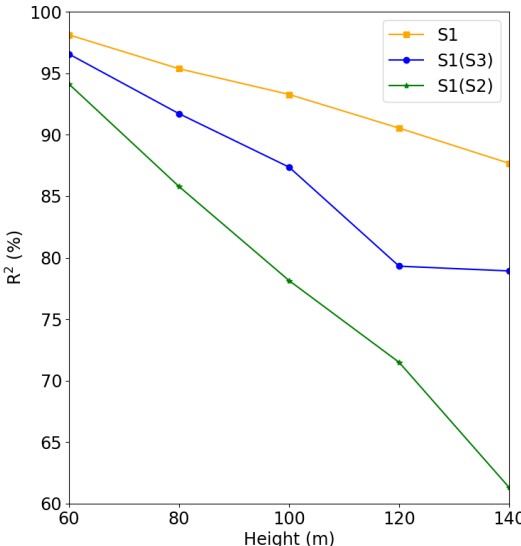

**Figure 13.** Comparison between the **Experiment 1** and **Experiment 3**, where S1 is the result of the Experiment 1; S1(S2) is the forecast for the Site 1 ran with the model of the Site 2; and S1(S3) is the forecast for the Site 1 ran with the model of the Site 3

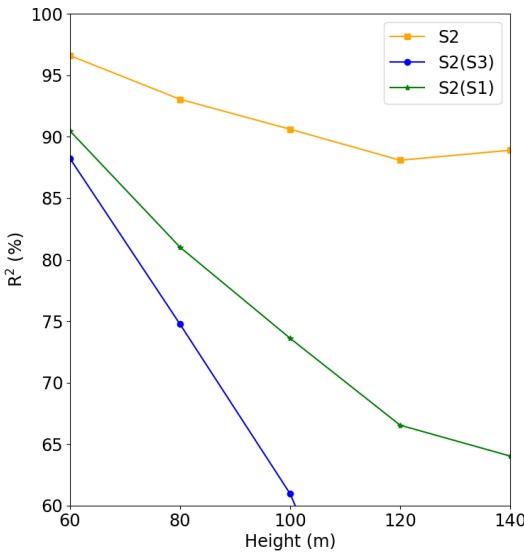

**Figure 14.** Comparison between the **Experiment 1** and **Experiment 3**, where S2 is the result of the Experiment 1; S2(S1) is the forecast for the Site 2 ran with the model of the Site 1; and S2(S3) is the forecast for the Site 2 ran with the model of the Site 3

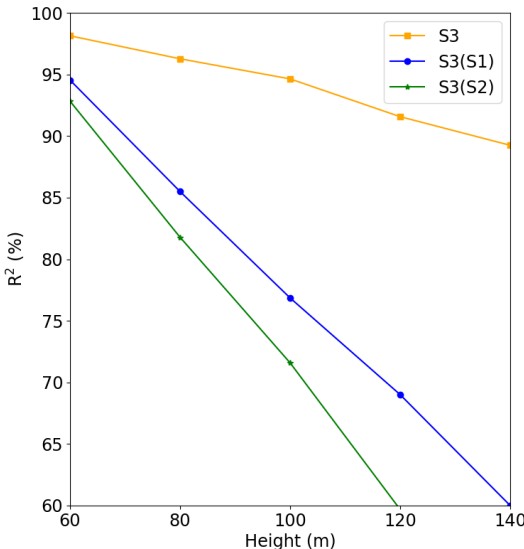

**Figure 15.** Comparison between the **Experiment 1** and **Experiment 3**, where S3 is the result of the Experiment 1; S3(S1) is the forecast for the Site 3 ran with the model of the Site 1; and S3(S2) is the forecast for the Site 3 ran with the model of the Site 1

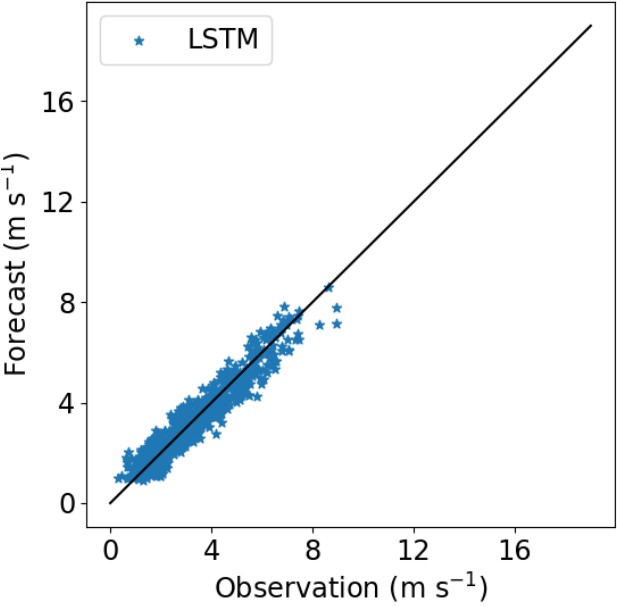

**Figure 16.** Correlation between forecasted and observed data for the Site 1 with the model trained at Site 3. Height: 80 m

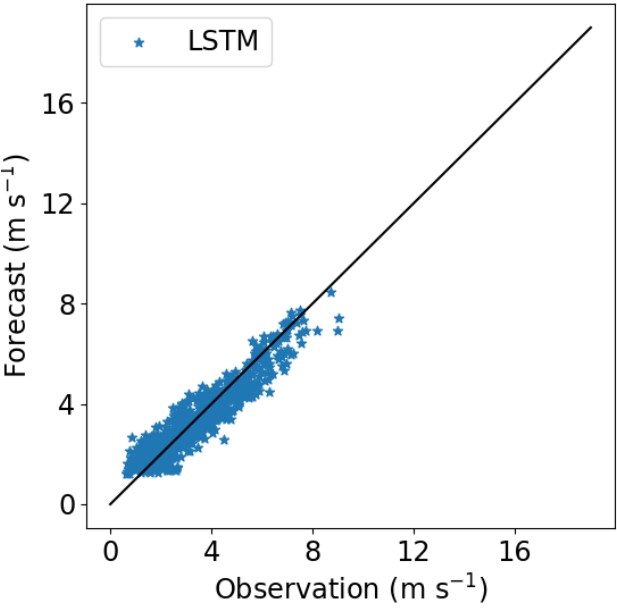

**Figure 17.** Correlation between forecasted and observed data for the Site 1 with the model trained at Site 3. Height: 100 m

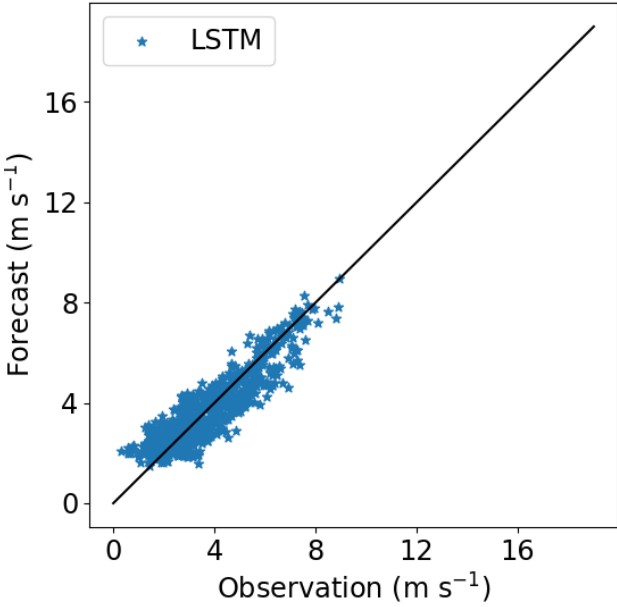

**Figure 18.** Correlation between forecasted and observed data for the Site 1 with the model trained at Site 3. Height: 140 m

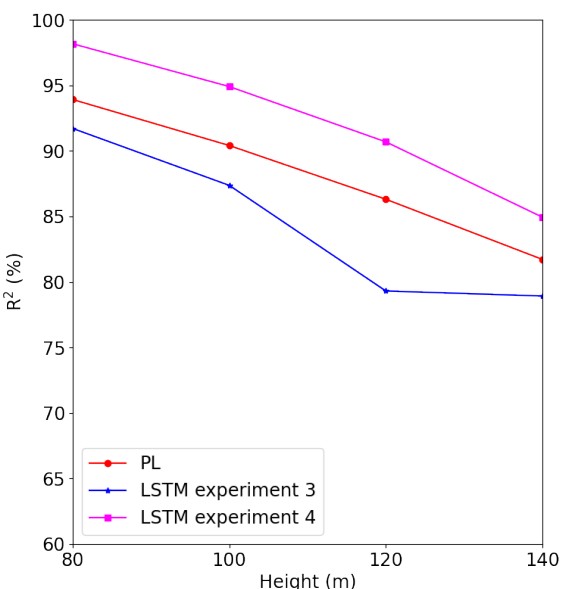

**Figure 19.** Site 1: Comparison for the **PL**, **Experiment 3** and **Experiment 4** estimates

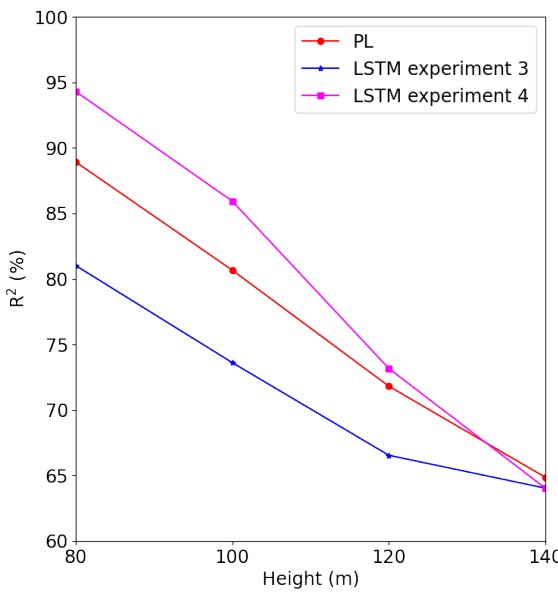

**Figure 20.** Site 2: Comparison for the **PL**, **Experiment 3** and **Experiment 4** estimates

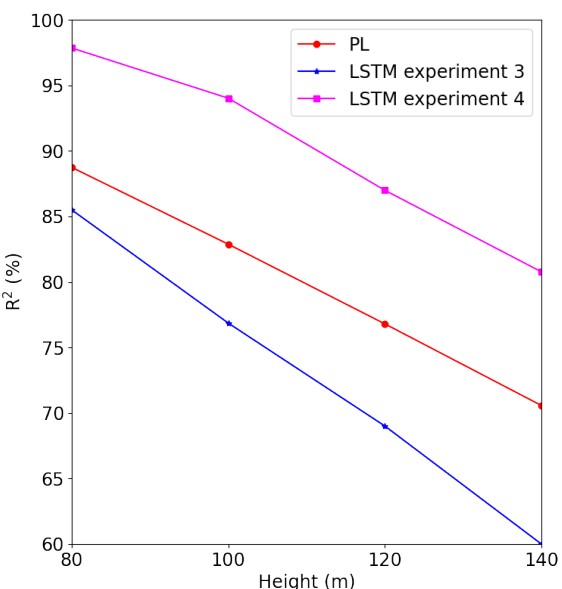

**Figure 21.** Site 3: Comparison for the **PL**, **Experiment 3** and **Experiment 4** estimates

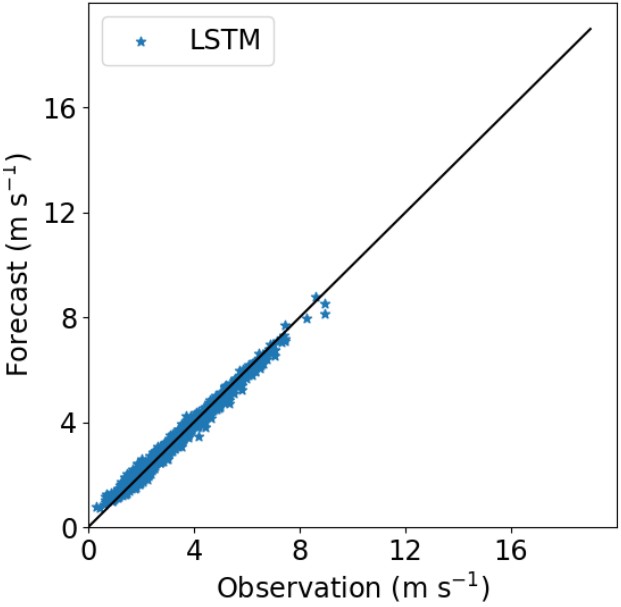

**Figure 22.** As Fig. 16, except the 60 m wind speed was added to the input dataset

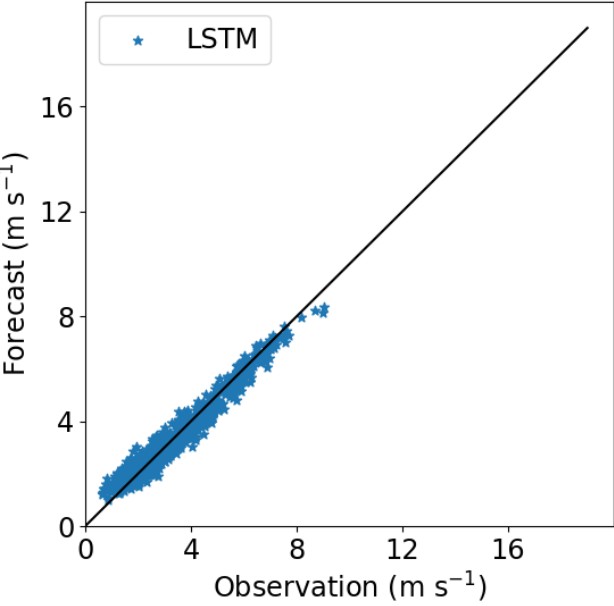

**Figure 23.** As Fig. 17, except the 60 m wind speed was added to the input dataset

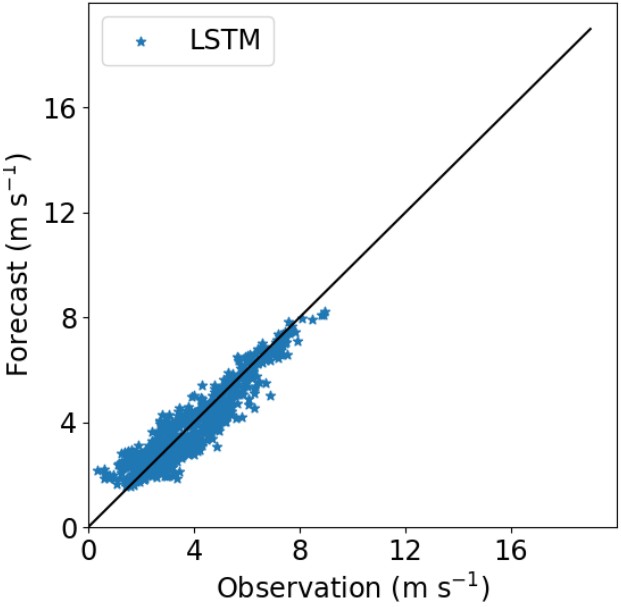

**Figure 24.** As Fig. 18, except the 60 m wind speed was added to the input dataset

## 4 Conclusions

Nowadays, the machine learning techniques produce successful results to forecast environmental processes. However, forecasting the wind speed is still a challenge due its random nature and researchers are dedicating considerable time and efforts to reach confident results. Comparative studies showed the superiority of the LSTM to forecast the wind speed against other machine learning techniques. Adding more meteorological variables has also improved the results. Ensemble and hybrid methods are strategies that also contribute to the model performances.

Only recently, the machine learning techniques have been applied to extrapolate the wind speed to higher heights. The models generally require large datasets with some observational heights. After testing some commonly used algorithms for the wind speed forecast (Random Forest Trees, Support Vector Regression and Multi-layer Perceptron), we found out the LSTM outperformed all of them. The LSTM outperformed even the decomposition methods.

We also evaluated different dataset sizes and found out that the model didn´t improve even if the dataset size increases beyond that presented in Table A1; however, the model is sensitive to the training data percentage. In this study, taking 90% of the dataset for training produced the best result. The tests also showed best results for 10-min mean as input data than for 30-min or 1-hour mean.

Including the 40 m wind direction, TKE and the hour to the input dataset improved the model, which outperformed the Power Law as the distance from the surface increases. Adding the 60 m wind speed observations to the dataset improved the results, as expected from results of previous studies. However, the improvement was better to the Sites 1 and 2 than for the Site 3. The causes should be investigated in a future work.

Even over complex terrain and with a relatively short dataset (an observational campaign shorter than 3 months), the LSTM outperformed the Power Law. The Power Law cannot reproduce features like the LLJs that are often observed, at least over the Sites 1 and 3. The Site 2 is strongly influenced by the sea and land breezes and the LSTM model captured the abrupt changes of the wind profile better than the Power Law.

The results found from this observational campaign, albeit short, show the benefits of Doppler Lidar in improving model results to estimate winds at height. This is particularly relevant to help support the energy transition and net zero targets. Despite the costs associated with Doppler Lidars, the authors would encourage further strategic collaborations to drive observational data improvements leading to advances in model prediction.

As future work, we intend to follow two different approaches. As we took better results with 10 min mean than with 1 hour mean observational data, we want to test the LSTM recurrent neural network providing higher temporal resolution dataset, like 1 min or 5 min means, instead of 10 min means, as provided to this study. Even this requires a new observational campaign, we could evaluate the benefits from increase the temporal resolution of the dataset. As second approach we would like to evaluate the benefits of adding other sources, like reanalysis data or more observational data, as surface pressure, surface temperature, 2 m and 10 m wind data. Concerning to machine learning techniques, we suggest deeper investigation through hybrid and ensemble methods.

*Code and data availability.* For the LSTM model design see https://github.com/cassiabeu/doi.org-10.5194-wes-2023-104.git. Datasets are available under request. Please, contact cassia.beu@gmail.com

 **Appendix A**

**Table A1.** Dataset size and hyperparameters

| site | data points | Training data (%) | Experiment 1 | | | Experiment 2 | | |
| --- | --- | --- | --- | --- | --- | --- | --- | --- |
| | | | units | epochs | batch size | units | epochs | batch size |
| 1 | 10000 | 90 | 30 | 30 | 2 | 30 | 30 | 2 |
| 2 | 12000 | 90 | 20 | 20 | 2 | 15 | 30 | 2 |
| 3 | 8000 | 90 | 50 | 150 | 2 | 20 | 70 | 2 |

**Table A2.** LSTM arguments

| Argument | Value |
|---|---|
| **activation** | "tanh" |
| **recurrent_activation** | "sigmoid" |
| **use_bias** | True |
| **kernel_initializer** | "glorot_uniform" |
| **recurrent_initializer** | "orthogonal" |
| **bias_initializer** | "zeros" |
| **unit_forget _bias** | True |
| **kernel_regulatizer** | None |
| **bias_regulatizer** | None |
| **activity_regulatizer** | None |
| **kernel_constraint** | None |
| **recurrent_constraint** | None |
| **bias_constraint** | None |
| **dropout** | None |
| **recurrent_dropout** | 0.0 |
| **seed** | 0.0 |
| **return_sequences** | None |
| **return_state** | False |
| **go_backwards** | False |
| **statetul** | False |
| **unroll** | False |

**Table A3.** Data availability (%)

| Height (m) | Site 1 | Site 2 | Site 3 |
|:---:|:---:|:---:|:---:|
| **40** | 100 | 100 | 100 |
| **60** | 100 | 100 | 100 |
| **80** | 100 | 100 | 100 |
| **100** | 100 | 100 | 100 |
| **120** | 100 | 100 | 100 |
| **140** | 100 | 100 | 100 |
| **160** | 100 | 100 | 99 |
| **180** | 100 | 99 | 98 |
| **200** | 100 | 99 | 97 |
| **230** | 100 | 98 | 94 |
| **260** | 100 | 95 | 90 |
| **290** | 100 | 92 | 83 |

**Table A4.** Input variables

| Experiment | Variables |
|:---:|:---:|
| **1** | hour, 40 m wind speed, 40 m wind direction, 40 m TKE |
| **2** | hour, 40 m wind speed, 40 m wind direction, 40 m TKE, 60 m wind speed |
| **3** | hour, 40 m wind speed, 40 m wind direction, 40 m TKE |
| **4** | hour, 40 m wind speed, 40 m wind direction, 40 m TKE, 60 m wind speed |

**Appendix B**

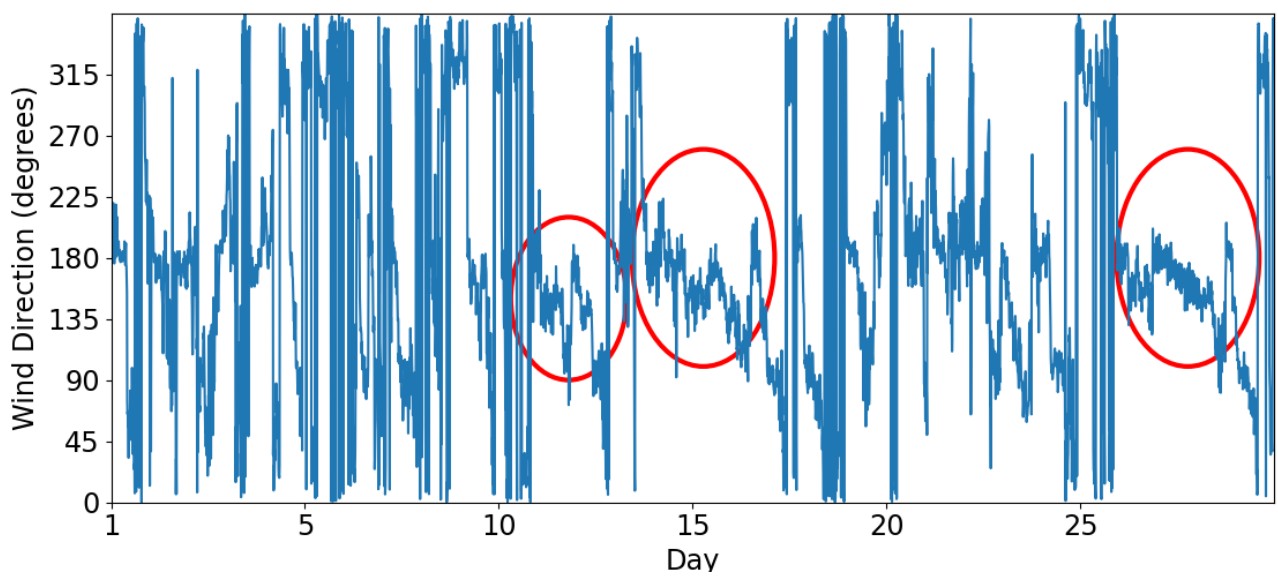

**Figure A1.** Site 1: Wind direction temporal series

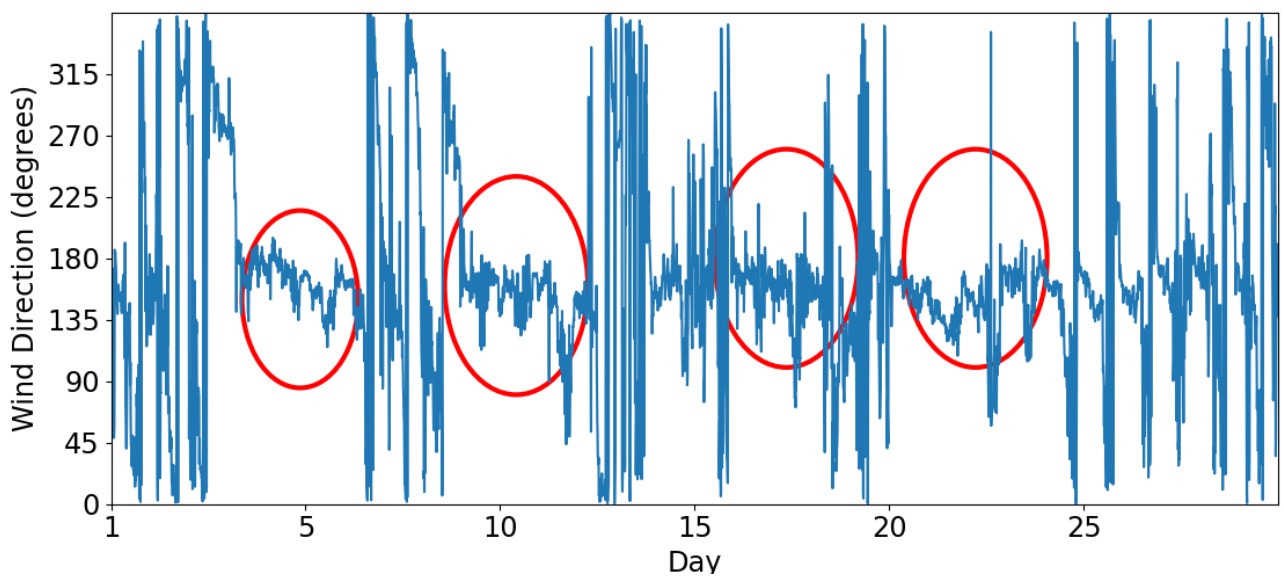

**Figure A2.** Site 2: Wind direction temporal series

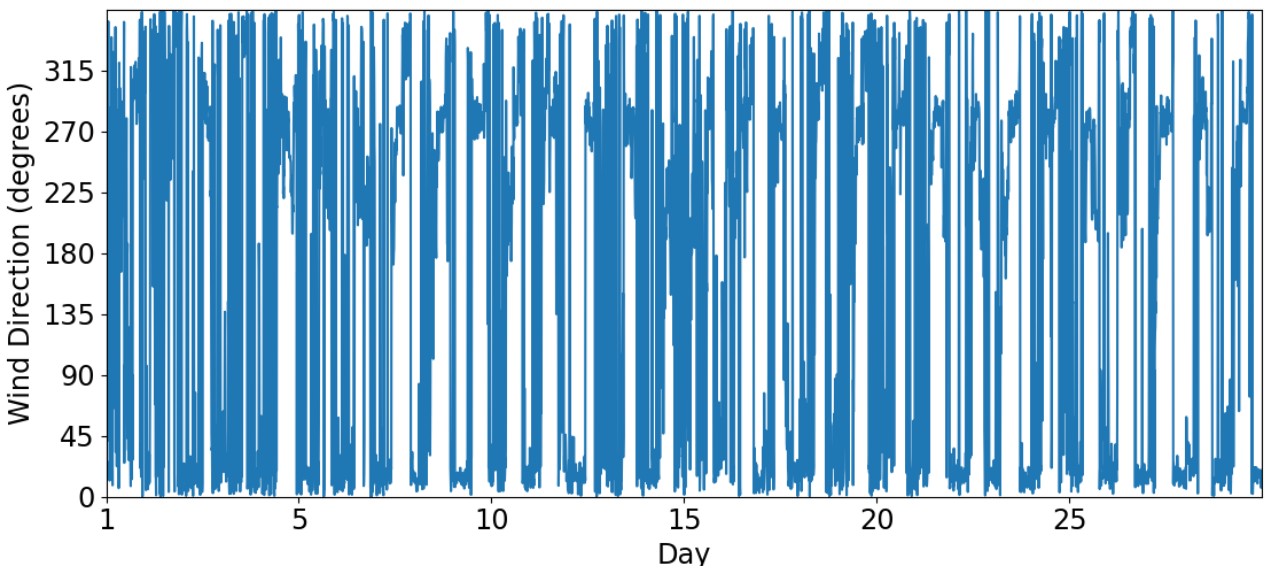

**Figure A3.** Site 3: Wind direction temporal series

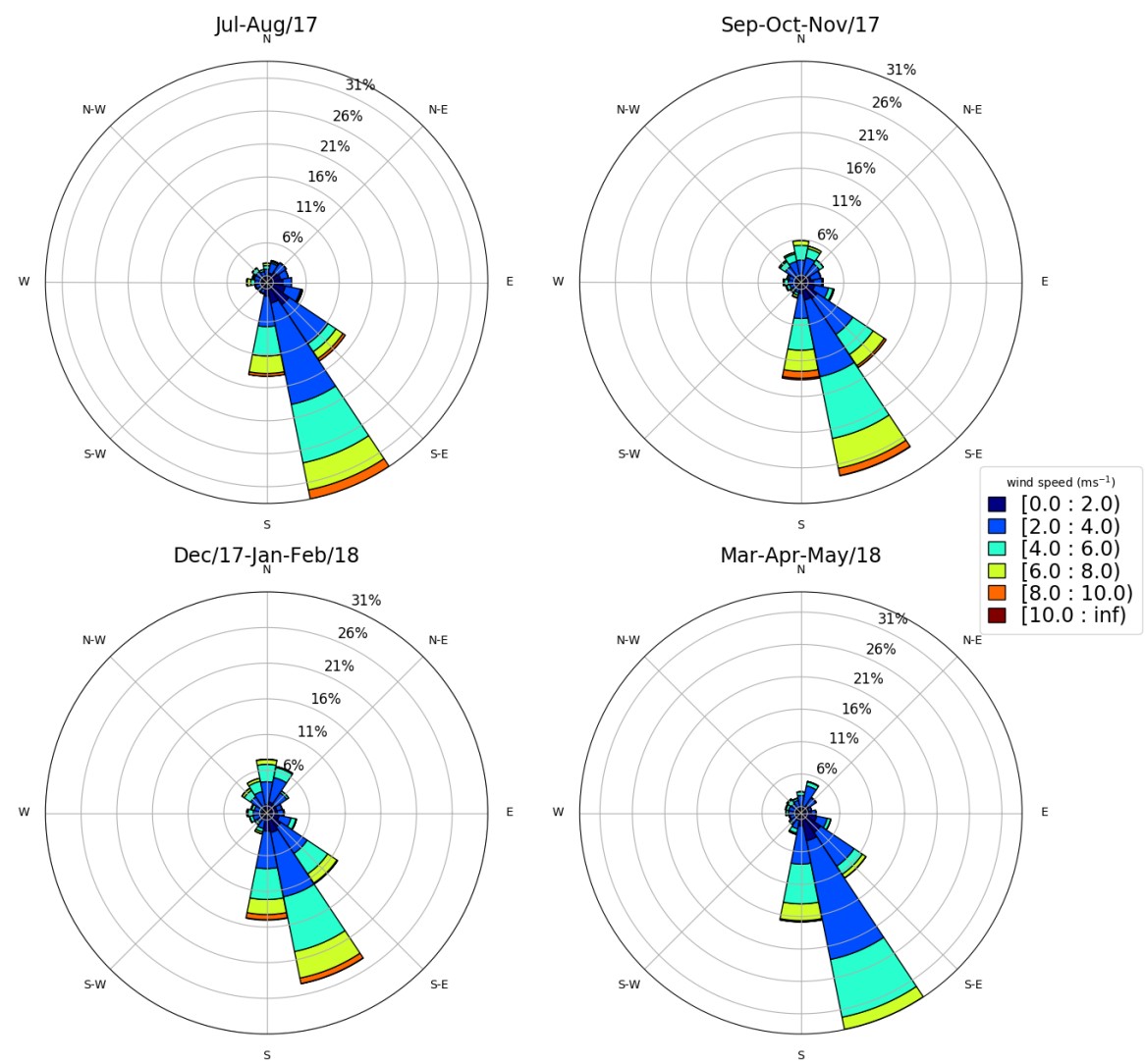

**Figure A4.** Site 3: Observed wind at 40 m (normalized)

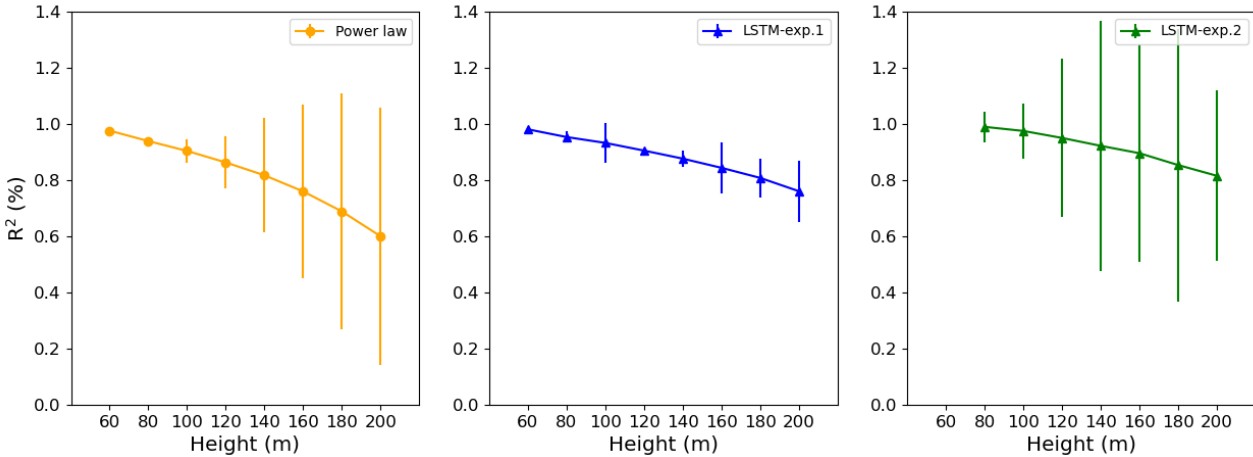

**Figure A5.** Site 1: $R^2$ error bars

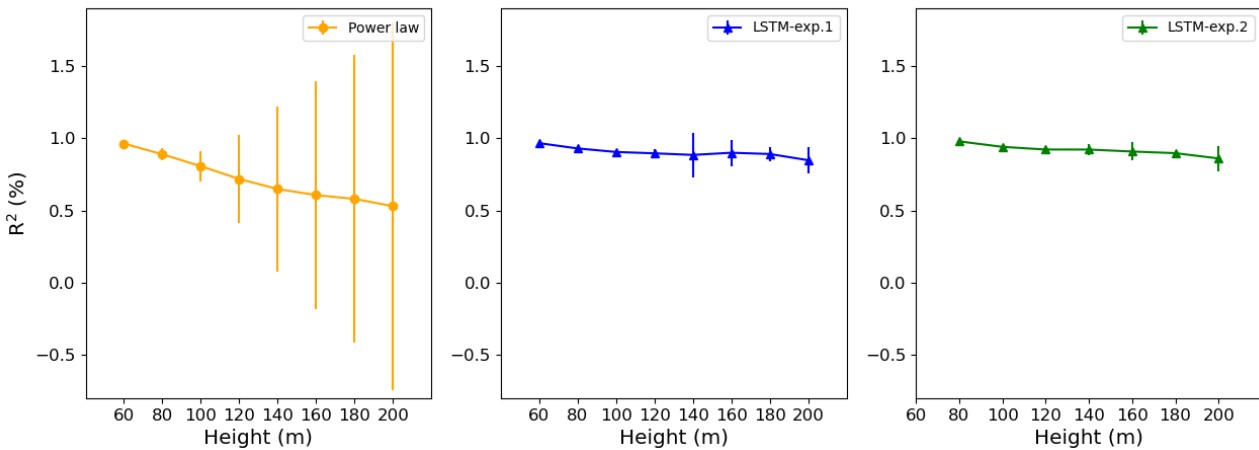

**Figure A6.** Site 2: $R^2$ error bars

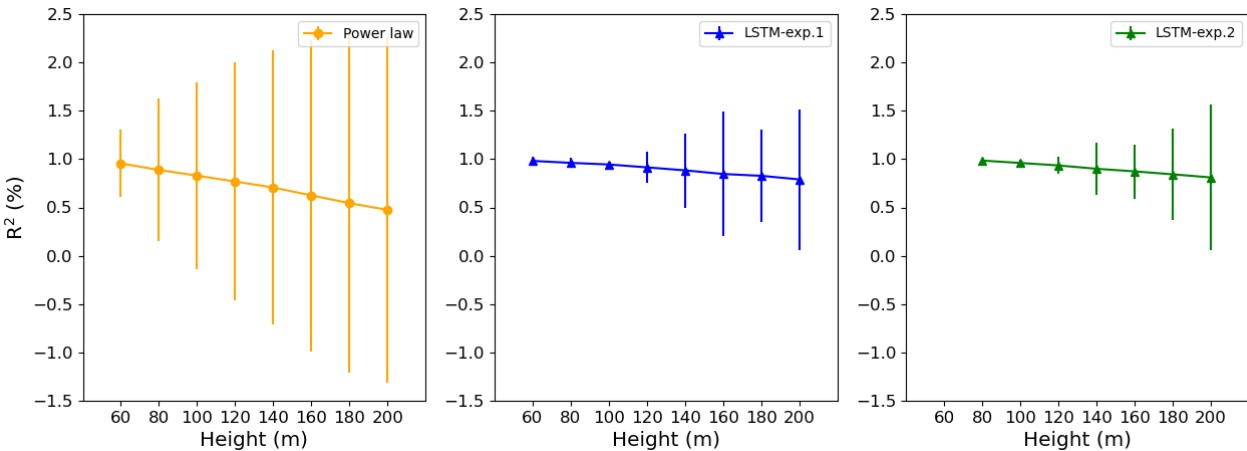

**Figure A7.** Site 3: $R^2$ error bars

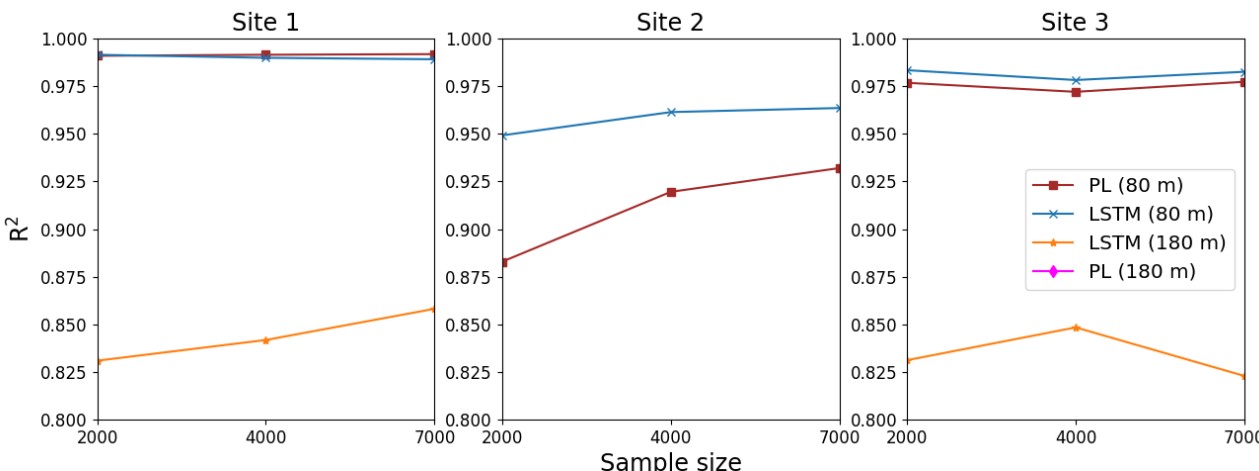

**Figure A8.** $R^2$ versus sample size of the **Experiment 2** tests. $\alpha$ estimated from observational 40 and 60 m wind speed

*Author contributions.* The authors contributed equally to this work.

*Competing interests.* There are no competing of interests.

*Acknowledgements.* The authors acknowledge the Instituto Nacional de Pesquisas Energéticas (IPEN) and the Financiadora de Estudos e
Projetos (FINEP).

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
