# Peer review of "Machine Learning-Based Estimate of The Wind Speed Over Complex Terrain Using the LSTM Recurrent Neural Network"

_Wind Energy Science, 2023_

## Referee Comment (RC1)

**General Comments**

This article works to further the use and understanding of wind forecasting using machine learning algorithms. This work uses wind observations at lower levels to make timeseries predictions of winds higher above the surface and compares them to the established method, the Power Law. The authors chose locations with complex terrain to experiment with predictive methods to research where they have the greatest forecast skill. They also tested the viability of swapping models to different environments to look at their fit under different conditions.

The authors break down the model output as a function of the local environments at the different sites, taking the time to delve into the meteorological reasons why the outputs look the way that they do. Despite this, I have concerns about the application of the Power Law, with α being kept constant. As alpha is explained to be a proxy for atmospheric stability, the different terrain scenarios will greatly affect this (i.e. stability is rapidly affected by the onset and movement of the sea breeze and land breeze), especially because the replication of the diurnal cycle was important to the results. The period of record in the training datasets will likely not capture the environmental response to the changing of seasons. The article opens with a lot of connections to previous works but fails to discuss any connections between the research done here and those previous studies.

Most of the paper requires grammatical revisions. There are also portions of the manuscript that do not pertain to the subject matter at hand. This work tackles research that can benefit the scientific community and the furthering of wind forecasting and its application with machine learning methods and their portability, thus I would recommend significant revisions.

**Specific Questions/Comments**

*Page 1, Lines 14-24:* The first paragraph of the introduction lacks substance that is specific to the work being done here. It discusses the use of Machine learning and its applications, but the scope is too broad. This may be done more concisely by collapsing this paragraph into a sentence or two discussing the variety of machine learning applications with several examples, while transitioning quickly into your second paragraph, which is far more relevant.

*Page 4, Lines 88-94:* Along with the governing equations of the LSTM cell, I suggest adding a diagram to help the reader. You can also connect to the different elements within the cell to strength your further explanation on lines 95-97.

*Page 5, Line 122:* The explanation for Experiment 3 and 4 needs more clarity. After reading into the results, I understood that your goal was to apply each trained model to a site that it was not originally trained for, however this was not clear when it was first proposed.

*Page 5&6, Section 2.4:* You lead off with the statement of "we chose the metrics that have typically been used in similar works". This statement is rather vague and could use some support. At the end of the paragraph, you say "Zhou et al. (2022) and Baquero et al. (2022) provide detailed explanations for those metrics". I suggest moving these references to the beginning of the paragraph and combining your statements… example: "We chose verification metrics like those used in Zhou et al. and Baquero et al., because…. For further information on these metrics, see Zhou and Baquero."

*Page 6, Line 150:* [This is a stylistic critique] You use the wording "a Metropolitan Region (São Paulo city)". Would it be better to say "the Metropolitan Region of São Paulo"? As the reader passes over parenthetical statements, they tend to cause a pause. This style of adding parenthetical statements for further description is used frequently in the paper. In my opinion, the number of parenthetical statements hurts the reading fluency and detracts from the reading experience. Please apply the stylistic switch where you deem necessary.

*Page 6, Lines 157-166:* The history of research on pollution in and around the Cubatão area is not greatly relevant to the background of this paper, nor its focus on wind forecasting. If the authors are trying to tie the importance of windspeed forecasting to the distribution of pollutants, this does not come across as a motivating factor clearly. If the authors did not intend for this to be a point of motivation for this research, I recommend the removal of this block of text.

*Page 7, Lines 170-171:* "Gasparoto et al. (2014) summarized some characteristics of the Ipanema National Forest." If there is further information that the reader should be aware of regarding the characteristics of the Ipanema National Forest that are relevant to the research being conducted at site 3, I recommend paraphrasing from the source provided or removing the sentence if the information is not overly relevant.

*Page 7, Line 179:* "Both of them present a diurnal cycle" Your figures do not show a time series showing this result. Although I would caution adding more figures, you may put some thought into showing a time series here.

*Page 8, Line 181:* Could you elaborate on impact of the LLJ at the heights above ground you are using in this research?

*Page 8, Lines 184 and 185:* [Klockow and Targa (1998) figure 2] The figure was *extremely* simplified, to the point that I was left wondering how it provided any insight into windspeed prediction or anything to do with the sea breeze. Could you please add further insight from the use of this referenced figure?

*Page 10, Lines 198-202:* There are several references made in these lines, but they are not referencing the papers which foundationally developed the models. I would recommend adding a reference to the foundational works in addition to the references that guided the research done here thru the application and review of these models.

*Page 10, Line 206:* "Thus, using the entire dataset for training and testing the model takes a while." Wording could be more professional here. As a technical comment, this is a good point to reference your computing resources. The use of Google Colab would obviously be different than if someone was training on new GPUs and a dedicated machine. The simplicity of this setup can be a benefit to further implementation on a wider scale by more users, as discussed on lines 294-297.

*Page 11, Line 207:* "At each step" At what steps? For each experiment?

*Page 11, Line 211:* "a short observational campaign can produce reliable results" Based on Table 1, your datasets only spanned certain times of the year sometimes missing seasons, and for site 3… entirely in the southern hemisphere winter. Is there any concern for the model's usability for prediction outside of these timeframes?

*Page 12, Lines 219-224:* This paragraph covers a lot of different results. Due to the number of figures and tables, many of them are referenced in this paragraph in quick succession. This made things very difficult to keep track of. I suggest spacing out references and/or adding more elaboration on the results from each figure/table to increase the value of their inclusion.

*Page 17, Line 261:* "The forecast for the Site 1 highly improves when the 60 m wind speed is included on the input dataset for training the model at Site 3". Where is the point of origin of the 60 m wind dataset? This has been left a bit unclear. See the silly equation below that popped into my head as I read this sentence…

[Prediction for Site 3] = [Model trained at Site 1] + ([60m data from Site 3]? Or [60m data from Site 1]?)

Additionally, have you considered adding 10m/2m data to any of these tests? These heights are the generally the most common in modelling and observed networks. When talking about applying a pretrained model from another region [maybe with similar terrain], it may be worthwhile to test this concept with data that would be more readily available at a wider range of locations.

*Page 27, Lines 276-277:* "Ensemble and hybrid methods are strategies that also contribute to the model performances." Were these concepts tested in this study? If they were not, maybe this should be moved to an area discussing future work.

*Page 27, Lines 279-281:* "After testing some commonly used algorithms for the wind speed forecast (Random Forest Trees, Support Vector Regression and Multi-layer Perceptron), we found out the LSTM outperformed all of them. The LSTM outperformed even the decomposition methods." These models were briefly discussed on page 9 and 10 starting at the end of line 196, but no results were ever shown to the reader, nor was it said that those models had poorer performance, only that "From this point, we refer to results with the LSTM RNN". This needs to be revised in the results section before this statement is made in the conclusion.

*Page 27, Line 283:* "90%" How much of the remaining 10% of the data went towards validation and testing?

*Page 27, Line 288:* "However, the improvement was better to the Sites 1 and 2 than for the Site 3" Is there any speculation or further explanation of why this was the outcome?

*Page 27, Lines 292-293:* "The Site 2 is strongly influenced by the sea and land breezes and the LSTM model captured the abrupt changes of the wind profile better than the Power Law" Please see my major comment in the 'general comments' section about my concerns with keeping alpha constant when implementing the Power Law. This is one of the cases where there would be a rapid change in stability. You incorporated 60m data into the training of your models, so can you use the change in windspeed between 40-60m over a short timeseries to better estimate alpha ~windshear coefficient?

*Page 27, Lines 294-297:* Reword these statements for better clarity.

**Technical Corrections**

Quick preface, if something is corrected here, it may have occurred more than once, however I will only be pointing out the first instance.

*Page 1, Line 6:* The use of the word "until" in this spot is grammatically incorrect. "Up to" may be a valid correction.

*Page 1, Line 8:* "variables of 40 m" changed to "variables at 40 m".

*Page 2, Lines 39-40:* "Due the random feature of the wind speed" recommended rewording "Due to the random nature of windspeed when looking at data over short time scales".

*Page 3, Lines 81-82:* "state-of-art" changed to "state-of-the-art".

*Page 4, Table 1:* "31/Dec/2-16" changed to "31/Dec/2016"

*Page 4, Lines 106-107:* "retrieve information each 10 minutes" changed to "retrieve information every 10 minutes".

*Page 5, Lines 110-111:* "the standard deviation of the horizontal (σu + σv) e vertical (σw) wind speed to forecast the wind speed at higher heights" Please revise, I am unsure what 'e vertical' is referring to.

*Page 6, Line 151:* "the strong social difference" I am unsure what is meant here; maybe 'strong social disparity'?

*Page 6, Line 157:* "looks like a big wall." Please revise with more scientific language or remove this statement.

*Page 8, Figure 2:* There are numbers along the concentric circles of the wind rose that lack units. Based on other examples in literature, these are "%" percentages based on the normalized time period, i.e. relative frequency. See the crudely circled region of the plot shown below. Further explanation should be added to the caption accordingly.

[Figure]

*Page 9, Figure 3:* Values discussed in my comment above are overlapped by data, making the relative frequency values impossible to read.

[Figure]

*Page 9, Line 190:* "along the day" change to "throughout the day"

*Page 9, Line 196:* "until to reach the results that will be presented in this section" Please revise wording. Maybe? "the model configurations that had the best performance will be presented in this section"

*Page 10, Line 205:* "50 thousand data" 50 thousand timesteps? Datapoints?

*Page 11, Lines 206-207:* "Surprisingly, we found that the model only improves until a limited dataset size and was unnecessary to take the entire dataset." Suggested edit for clarity "Surprisingly, we found that model improvement plateaued without using all of the datapoints of record."

*Page 13, Line 238:* The references used here should be prefaced as examples. ~ (e.g. Vassallo et al., 2020; Mohandes and Rehman, 2018)

*Page 16, Figure 6:* The line representing the Power Law stops being visible around 170m along the x-axis. This would be more representative if we could see the behavior of all three models tested.

*Page 27, Lines 273-275:* Grammar issues in first two sentences of the conclusion. Please revise for clarity.

---

## Author Comment (AC1)

**General Comments**

This article works to further the use and understanding of wind forecasting using machine learning algorithms. This work uses wind observations at lower levels to make timeseries predictions of winds higher above the surface and compares them to the established method, the Power Law. The authors chose locations with complex terrain to experiment with predictive methods to research where they have the greatest forecast skill. They also tested the viability of swapping models to different environments to look at their fit under different conditions.

The authors break down the model output as a function of the local environments at the different sites, taking the time to delve into the meteorological reasons why the outputs look the way that they do. Despite this, I have concerns about the application of the Power Law, with α being kept constant. As alpha is explained to be a proxy for atmospheric stability, the different terrain scenarios will greatly affect this (i.e. stability is rapidly affected by the onset and movement of the sea breeze and land breeze), especially because the replication of the diurnal cycle was important to the results. The period of record in the training datasets will likely not capture the environmental response to the changing of seasons. The article opens with a lot of connections to previous works but fails to discuss any connections between the research done here and those previous studies.

Most of the paper requires grammatical revisions. There are also portions of the manuscript that do not pertain to the subject matter at hand. This work tackles research that can benefit the scientific community and the furthering of wind forecasting and its application with machine learning methods and their portability, thus I would recommend significant revisions.

[Figure]
 Dear referee,we acknowledge your comments. All the comments will surely enrich our work. We tried, on the best way, to attend all the contributions of both referees. Below, all the comments are answered. Concerning the grammar, if the draft is accepted, before publication, it will undergo a professional revision.

**Specific Questions/Comments**

*Page 1, Lines 14-24:* The first paragraph of the introduction lacks substance that is specific to the work being done here. It discusses the use of Machine learning and its applications, but the scope is too broad. This may be done more concisely by collapsing this paragraph into a sentence or two discussing the variety of machine learning applications with several examples, while transitioning quickly into your second paragraph, which is far more relevant.

[Figure]
. Done

*Page 4, Lines 88-94:* Along with the governing equations of the LSTM cell, I suggest adding a diagram to help the reader. You can also connect to the different elements within the cell to strength your further explanation on lines 95-97.

A. Done

*Page 5, Line 122:* The explanation for Experiment 3 and 4 needs more clarity. After reading into the results, I understood that your goal was to apply each trained model to a

site that it was not originally trained for, however this was not clear when it was first proposed.

A. Rewritten

*Page 5&6, Section 2.4:* You lead off with the statement of "we chose the metrics that have typically been used in similar works". This statement is rather vague and could use some support. At the end of the paragraph, you say "Zhou et al. (2022) and Baquero et al. (2022) provide detailed explanations for those metrics". I suggest moving these references to the beginning of the paragraph and combining your statements… example: "We chose verification metrics like those used in Zhou et al. and Baquero et al., because…. For further information on these metrics, see Zhou and Baquero."

A. Rewritten

*Page 6, Line 150:* [This is a stylistic critique] You use the wording "a Metropolitan Region (São Paulo city)". Would it be better to say "the Metropolitan Region of São Paulo"? As the reader passes over parenthetical statements, they tend to cause a pause. This style of adding parenthetical statements for further description is used frequently in the paper. In my opinion, the number of parenthetical statements hurts the reading fluency and detracts from the reading experience. Please apply the stylistic switch where you deem necessary.

A. Rewritten

*Page 6, Lines 157-166:* The history of research on pollution in and around the Cubatão area is not greatly relevant to the background of this paper, nor its focus on wind forecasting. If the authors are trying to tie the importance of windspeed forecasting to the distribution of pollutants, this does not come across as a motivating factor clearly. If the authors did not intend for this to be a point of motivation for this research, I recommend the removal of this block of text.

A. Rewritten

*Page 7, Lines 170-171:* "Gasparoto et al. (2014) summarized some characteristics of the Ipanema National Forest." If there is further information that the reader should be aware of regarding the characteristics of the Ipanema National Forest that are relevant to the research being conducted at site 3, I recommend paraphrasing from the source provided or removing the sentence if the information is not overly relevant.

A. Done

*Page 7, Line 179:* "Both of them present a diurnal cycle" Your figures do not show a time series showing this result. Although I would caution adding more figures, you may put some thought into showing a time series here.

A. Time series were added as appendices

*Page 8, Line 181:* Could you elaborate on impact of the LLJ at the heights above ground you are using in this research?

A. rewritten

*Page 8, Lines 184 and 185:* [Klockow and Targa (1998) figure 2] The figure was *extremely* simplified, to the point that I was left wondering how it provided any insight into windspeed prediction or anything to do with the sea breeze. Could you please add further insight from the use of this referenced figure?

> A. We agree that this a simplified conceptual model, but that was an important project in the sense of the first observational experiment including international cooperation. The intention of this reference is to illustrate the local circulation  and its complexity due to the sea-land contrast and orography based on a scientific publication. This complexity may prejudice the model performance

*Page 10, Lines 198-202:* There are several references made in these lines, but they are not referencing the papers which foundationally developed the models. I would recommend adding a reference to the foundational works in addition to the references that guided the research done here thru the application and review of these models.

> A. Done

*Page 10, Line 206:* "Thus, using the entire dataset for training and testing the model takes a while." Wording could be more professional here. As a technical comment, this is a good point to reference your computing resources. The use of Google Colab would obviously be different than if someone was training on new GPUs and a dedicated machine. The simplicity of this setup can be a benefit to further implementation on a wider scale by more users, as discussed on lines 294-297.

> A. Rewritten

*Page 11, Line 207:* "At each step" At what steps? For each experiment?

> A. Rewritten

*Page 11, Line 211:* "a short observational campaign can produce reliable results" Based on Table 1, your datasets only spanned certain times of the year sometimes missing seasons, and for site 3… entirely in the southern hemisphere winter. Is there any concern for the model's usability for prediction outside of these timeframes?

> A. Yes, there is concern about the model's usability. Regarding this, we analyzed different seasons for the Site 3, which had the longest continuous campaign, covering a whole year, between 2017 and 2018. Those results weren't reported on the text, but we identified only small variations compared to the table 3. Despite we saw a reduction of the correlation, generally the LSTM still performed better than the PL for 2 different periods (December – January and March – April). For the June – July period, the methods converge (the $R^2$ tend to same value). The tests were done for 60, 80 and 100 m.
>
> We believe that the success of these results stem from the LSTM abilities, as a RNN, to built a memory of time series.

*Page 12, Lines 219-224:* This paragraph covers a lot of different results. Due to the number of figures and tables, many of them are referenced in this paragraph in quick succession. This made things very difficult to keep track of. I suggest spacing out references and/or adding more elaboration on the results from each figure/table to increase the value of their inclusion.

A. Rewritten

*Page 17, Line 261:* "The forecast for the Site 1 highly improves when the 60 m wind speed is included on the input dataset for training the model at Site 3". Where is the point of origin of the 60 m wind dataset? This has been left a bit unclear. See the silly equation below that popped into my head as I read this sentence…

[Prediction for Site 3] = [Model trained at Site 1] + ([60m data from Site 3]? Or [60m data from Site 1]?)

A. The point of origin of the 60 m wind is the dataset for training:

[Prediction for Site 3] = [Model trained at Site 1 (40m wind speed + 40m wind dir + 40m TKE  + 60m wind speed)

Additionally, have you considered adding 10m/2m data to any of these tests? These heights are the generally the most common in modelling and observed networks. When talking about applying a pretrained model from another region [maybe with similar terrain], it may be worthwhile to test this concept with data that would be more readily available at a wider range of locations.

A. We totally agree with the revisor concerning to modelling with 2m/10m data. Although these heights are the more commonly available, during our experiments, we haven´t simultaneous 2m/10m observations. As we believe it can improve the modelling, we suggest it as future work in the last paragraph of conclusions.

*Page 27, Lines 276-277:* "Ensemble and hybrid methods are strategies that also contribute to the model performances." Were these concepts tested in this study? If they were not, maybe this should be moved to an area discussing future work.

A. Yes, following Zhou et al. (2022), we tested the Complete Ensemble Empirical Mode Decomposition with Adaptative Noise (CEEMDAN) method. Zhou et al. (2022) applied the CEEMDAN to forecast the carbon price, which is also a random variable. According to Zhou et al. (2022), decomposition-integration methods increase the accuracy and save time when compared with other hybrid methods and the CEEMDAN presents many advantages over its precursors. Those reasons motivated us to choose CEEMDAN over other methods.

Despite the CEEMDAN features, we hadn't as good result as we got if the LSTM. Anyway, we agree with the revisor that it still is a point for investigation and that the hybrid and ensemble methods should be tested deeper for better results. Suggestion included as future work.

*Page 27, Lines 279-281:* "After testing some commonly used algorithms for the wind speed forecast (Random Forest Trees, Support Vector Regression and Multi-layer Perceptron), we found out the LSTM outperformed all of them. The LSTM outperformed even the decomposition methods." These models were briefly discussed on page 9 and 10 starting at the end of line 196, but no results were ever shown to the reader, nor was it said that those models had poorer performance, only that "From this point, we refer to results with the LSTM RNN". This needs to be revised in the results section before this statement is made in the conclusion.

> A. Rewritten

*Page 27, Line 283:* "90%" How much of the remaining 10% of the data went towards validation and testing?

> A. All the remaining 10% of the data were used for validation. Then, for testing the models we took more data. We included the explanation in the last paragraph of section 3.2 - Experiment 2.

*Page 27, Line 288:* "However, the improvement was better to the Sites 1 and 2 than for the Site 3" Is there any speculation or further explanation of why this was the outcome?

> A. At this moment we don't have further explanation for that behavior. Maybe more Planetary Boundary Layer analyses could help us understand those differences.

*Page 27, Lines 292-293:* "The Site 2 is strongly influenced by the sea and land breezes and the LSTM model captured the abrupt changes of the wind profile better than the Power Law" Please see my major comment in the 'general comments' section about my concerns with keeping alpha constant when implementing the Power Law. This is one of the cases where there would be a rapid change in stability. You incorporated 60m data into the training of your models, so can you use the change in windspeed between 40-60m over a short timeseries to better estimate alpha ~windshear coefficient?

> A. Suggestion accepted. The windshear coefficient was estimated and results are discussed in the last paragraph of Section 3.2 and Figure A5.

*Page 27, Lines 294-297:* Reword these statements for better clarity.

> A. Done

**Technical Corrections**

Quick preface, if something is corrected here, it may have occurred more than once, however I will only be pointing out the first instance.

*Page 1, Line 6:* The use of the word "until" in this spot is grammatically incorrect. "Up to" may be a valid correction.

> A. Done

*Page 1, Line 8:* "variables of 40 m" changed to "variables at 40 m".

> A. Done

*Page 2, Lines 39-40:* "Due the random feature of the wind speed" recommended rewording "Due to the random nature of windspeed when looking at data over short time scales".

> A. Done

*Page 3, Lines 81-82:* "state-of-art" changed to "state-of-the-art".

> A. Done

*Page 4, Table 1:* "31/Dec/2-16" changed to "31/Dec/2016"

> A. Done

*Page 4, Lines 106-107:* "retrieve information each 10 minutes" changed to "retrieve information every 10 minutes".

> A. Done

*Page 5, Lines 110-111:* "the standard deviation of the horizontal ($\sigma u + \sigma v$) e vertical ($\sigma w$) wind speed to forecast the wind speed at higher heights" Please revise, I am unsure what 'e vertical' is referring to.

> A. Done

*Page 6, Line 151:* "the strong social difference" I am unsure what is meant here; maybe 'strong social disparity'?

> A. Done

*Page 6, Line 157:* "looks like a big wall." Please revise with more scientific language or remove this statement.

> A. Done

*Page 8, Figure 2:* There are numbers along the concentric circles of the wind rose that lack units. Based on other examples in literature, these are "%" percentages based on the normalized time period, i.e. relative frequency. See the pdf file for a screenshot that depicts the content of this comment.

> A. The wind roses were corrected and now include the "%" signal.

*Page 9, Figure 3:* Values discussed in my comment above are overlapped by data, making the relative frequency values impossible to read. See pdf for the illustration of this comment.

> A. The wind roses were corrected and now the values are visible

*Page 9, Line 190:* "along the day" change to "throughout the day"

> A. Done

*Page 9, Line 196:* "until to reach the results that will be presented in this section" Please revise wording. Maybe? "the model configurations that had the best performance will be presented in this section"

   A. Done

*Page 10, Line 205:* "50 thousand data" 50 thousand timesteps? Datapoints?

   A. Done

*Page 11, Lines 206-207:* "Surprisingly, we found that the model only improves until a limited dataset size and was unnecessary to take the entire dataset." Suggested edit for clarity "Surprisingly, we found that model improvement plateaued without using all of the datapoints of record."

   A. Done

*Page 13, Line 238:* The references used here should be prefaced as examples. ~ (e.g. Vassallo et al., 2020; Mohandes and Rehman, 2018)

   A. Done

*Page 16, Figure 6:* The line representing the Power Law stops being visible around 170m along the x-axis. This would be more representative if we could see the behavior of all three models tested.

   A. That happened because we cut the y-axis ($R^2$) at 60% and the Power Law underperforms dramatically above 170 m compared to the other two models. Keeping the same resolution for the three sites eases the reading..

*Page 27, Lines 273-275:* Grammar issues in first two sentences of the conclusion. Please revise for clarity.

   A. Done

---

## Author Comment (AC2)

This is a relevant study for the Wind Energy community, presenting a machine learning approach to estimate wind profiles over complex terrain, using valuable input data from field campaigns in the global South (where observations are typically more sparse).

My comments and suggestions are intended to improve the clarity of the paper.

A. Dear referee, we acknowledge all your valuable comments. We are sure that it´s a worthy contribution to our work. We tried, on the best way to attend all the contributions of both referees. Following, all the comments are answered.

Minor Comments - language and typos:

Lines 6-7: Replace 'until' with 'up to'.

A. Done

Line 8: Replace ' when the input dataset included only variables of 40 m height' with 'for input variables up to 40 m high only'.

A. Done

Line 10: Replace 'others' with 'other'.

A. Done

Line 14: Replace 'is' with 'are'; and 'adopted as a power tool' with 'adopted as power tools'.

A. Done

Lines 38-40: These two sentence could be improved. Here's a suggestion: 'Wind forecasts underpin wind power prediction, which is essential to support  wind energy production in the short-term. Although winds have been traditionally forecasted with Numerical Weather Prediction models, the use of Machine Learning has become more widespread, not only to correct the biases derived from the highly variable nature of the winds, but also as stand-alone prediction models. Wang et al. (2021) ...'.

A. Done

Line 43: Replace 'some works' with ' a few studies'.

A. Done

Lines 44-45': Replace sentence as: 'They proposed the use of Long-Short Term Memory (LSTM) to improve wind speed forecasting for power prediction.'

A. Done

Lines 48-49: Replace 'and still recommend that the adaptiveness of the hybrid models needs to be further researched' with 'whilst recommending further investigation on the capabilities of hybrid model approaches'.

A. Done

Lines 53-54: Delete all 'the'.

A. Done

Line 62: Replace 'others' with 'other'.

A. Done

Line 64: Replace 'Following the same tendency' with 'Similarly, '.

A. Done

Line 66: Replace as 'and observations at four different heights as input.

A. Done

Lines 67-69: Replace sentence as: 'Bodini and Optis (2020a) and Bodini and Optis (2020b) found that random forests outperform standard wind extrapolation approaches, using a "round-robin" validation method. They highlighted the benefits of including observational data capturing the diurnal variability of the atmospheric boundary layer, namely the Obukhov length (...) at 4 m high.'.

A. Done

Lines 72- 73: Replace as: ', advising to carefully select the input data and emphasizing the importance of normalization'.

A. Done

Line 74: Replace 'over almost plain terrains' with 'over low complex terrain'.

A. Done

Line 75: Replace 'to' with 'in'; 'the most of the studies' with 'most studies'.

A. Done

Line 76: Replace 'for plain terrains' with 'low complexity orographies'; 'that analyzed different surfaces' with 'who analyzed different types of terrain complexity'.

A. Done

Line 79: Replace as: Recurrent Neural Networks (RNNs) are a type of artificial neural networks where the output of one time step is used as an input ...'.

A. Done

Line 109,120: Where you say 'levels' do you mean 'heights'?

A. Corrected

Line 111: Replace 'e' with 'and'.

A. Done

Lines 128-129: Replace start of the sentence: 'The authors state that:'.

A. Done

Lines 146: Replace 'occurred along three years' with 'took place over a three-year period'.

A. Done

Lines 146-147: Replace sentences as: 'All three observational sites are within x km from the coast, and clearly marked on the map.' - it'd be useful to know of far exactly they are from the coast, as proximity to the sea has a great influence over sea breezes and consequently the wind profile'.

A. Done

Line 148-149: Replace 'among each other' with 'between sites'; 'the surfaces' with 'types of terrain'; 'starting by the altitude ...' with 'namely the height and surface roughness'. - My understanding is that here you mean height (above ground) and not altitude (above mean sea level), although altitude is also a relevant factor to consider.

A. Done

Line 150-152: Replace 'by a chaotic constructive pattern that mixes high buildings common to a big city and simple residences' with 'by a densely mixed urban matrix'. (As is Sao Paulo, with a mixture of densely packed high-rise buildings and lower residential dwellings). I'd also remove the last sentence.

A. Done

Lines 153-154: Delete the first 'The'; '... which is characterized by a large number of industries from the largest harbor complex in Latin America.'.

A. Done

Line 155: Replace 'mountain chain ...' with 'mountain range, where Serra do Mar rises sharply, up to more than 700 m high, across 5 km wide.'.

A. Done

Line 167: Delete the first 'The'.; replace as ', as shown in Fig. 1.'

A. Done

Line 169: Replace as ' is the Aracoiaba Hill to the southeast, rising up more than 300 m high up to 900 m altitude.'

A. Done

Lines 195-222: Replace simply as: '... testing different machine learning models with multiple configurations, namely: Random Forests (...). Here we only present results for the best performing model LSTM RNN ...'.

A. Done. More references were included following the suggestions of the other Revisor.

Lines 204-205: Replace as: 'Data from Site 3 was first used to train the model; starting with wind speeds at 40 m to predict speeds at higher heights'.

A. Done

Line 209-210: Replace as: 'As the time series is comprised of 10-min temporal averages, that corresponds to roughly two months worth of observational data.'.

A. Done

Line 212: Delete 'The' before tables.

A. Done

Line 215: Replace 'As we see, for the three sites' with 'For all three sites'.

A. Done

Lines 294-297: Replace as: 'The results found from this observational campaign, albeit short, show the benefits of Doppler Lidar in improving model results to estimate winds at height. This is particularly relevant to help support the Energy transition and Net Zero targets. Despite the costs associated with Doppler Lidars, the authors would encourage further strategic collaborations to drive observational data improvements leading to advances in model prediction.'

A. Done

Major comments - literature review:

Lines 15-37: Although the literature review on Machine Learning in the environmental sciences is interesting, I am struggling to see how this is relevant to the main focus of

the paper (apart from the Cubatao context given in page 6). I'd recommend replacing these paragraphs with a more significant state-of-the-art summary, for example about Machine Learning applied to wind profile prediction, wind resource assessment, and wind power forecasting. Here are a few suggestions:

A. As suggested by the other referee, we reduced the literature review focusing mainly on wind forecasts.We acknowledge all those valuable recommendations, as we could cite them through the text.

Lines 75-76: Regarding the statement made here about vertical wind extrapolation studies, I'd refer the authors to these works looking at a range of terrain complexities, as well as Machine Learning methods: https://doi.org/10.1002/we.2013; https://doi.org/10.1016/j.egyai.2022.100209. I'd suggest a read through these papers and references to complement the Introduction.

A. Done

Lines 156-168: This is really interesting for socio-economic context and from a geographic point of view, but again I am struggling to see how it is relevant to the overarching theme of the paper. I think the authors could simply refer the reader to the Vieira and Gramani (2015) paper for further reading. Instead, I'd suggest to add a few notes on the climatic characteristics for that Sao Paulo region (where the 3 sites are located), as well as orographic complexity indices at each site (e.g. roughness length or silhouette area per unit area, if available). How are seasons typically classified for those areas and terrains - this would help complement the interpretation stated in page 9.

A. We removed part of the text, as suggested.

Major comments - graphics:

Lines 88-97: The mathematical explanation of the LSTM could benefit from a diagram of the network, to help readers understand visually the in/out/update mechanisms.

A. Done

Figures 2,3,4: Would it be possible to round the edges, so that they match the circle lines? Also, it'd be useful to see seasonal results for wind roses per month or season - these could be added to the appendix.

A. Those figures were redone, however, it was not possible to round the edges using the windrose library from Python. The seasonal wind roses were also included for the Site 3, because that Site has a continuous year of observational data.

Figures 5,6,7: Please add errors bars. Could you also add subplots for RMSE, MAE and F-score?

A. RMSE and MAE subplots were included with R2. As the error bars overlapped, we presented them on the appendix, as a subplot for each curve.

Regarding to F_score, this metric is mostly used for classification algorithms, while R2 is mostly used for regression.

Major comments - requiring additional input:

Lines 99-100: Can you please expand how the interpolation was performed and how the data was normalized? For example, was it a simple linear interpolation, or normalized by the maximum wind speed value? This is important to understand how the input data was processed (e.g. were diurnal cycle or spatial corrections accounted for) and how that could be affecting the representativeness of the outputs. It would also be useful to know the percentage of missing data that was processed in this way.

A. For the normalization we used the StandardScaler function from the Scikit Learn Library, as referenced (Pedregosa et al., 2011). This function standardizes the sample by removing the mean and scaling to unit variance. For a observed data (x), its standard value is:

z = (x - u)/s, where u is the mean and s is the standard deviation

The StandardScaler function was also applied by Schwegmann et al. 2023 (https://doi.org/10.1016/j.egyai.2022.100209)

The interpolation was done with the interpolate function from Pandas using the linear method.

These information were included in Section 2.1.

Table A3 exhibits the data availability.

Section 2.4 and Table 4: Can I suggest adding results for Root Mean Square Error (RMSE), so that error values are in the same scale as MAE and MAPE? Also, F-score is widely used in ML model evaluation and comparison - please consider computing it.

A. RMSE included.

Lines 206-208; 219-220; 225-226: Regarding the statement "Thus, using the entire dataset for training and testing the model takes a while. (...) was unnecessary to take the entire dataset" - how long is 'a while', and did you consider compute optimization options such as parallelisation? This could help understand why a shorter input dataset was better - was it just due to compute constraints, or was it the quality of the data? I'd second these questions for Sites 1 and 2.

A. To answer this question, we set the run environment to TPU (https://cloud.google.com/tpu/docs/intro-to-tpu?hl=pt-br#cloud_tpus_are_not_suited_to_the_following_workloads) and ran the Site 3 dataset, changing the data points. Each time we added 1000 data points, the execution time increased by 20%. We didn´t appliy compute optimization, but now we consider it´s as option for future works.

Line 209: Can you please clarify about the train-test-validation split ratios for each site experiment (i.e. was it just train-test or did you reserve some for validation)? Also could you highlight in the manuscript whether the LSTM model was trained and optimised on a site-basis, rather than for all 3 sites simultaneously?

A. As indicated in Table A1, 90% was used for training and 10% for validation. For the tests we took three more samples for each site with 2000, 4000 and 7000 data points. Results are shown in Fig. A8.

The model was trained and optimized on a site-basis and the hyperparameters for each one are indicated at Table A1.

Sections 3.1-3.4: A list (e.g. adding to Table A1) of all input variables for each experiment, as well as any relevant LSTM hyperparameters and network functions, would be useful to share.

A. We added Tables A2 and A4.

Lines 210-211: Regarding the statement: "This means that a short observational campaign can produce reliable results", I think it's a bit premature to reach that conclusion, without seeing results for at least a whole annual cycle. My recommendation is to delete this.

A. Removed

Lines 298-300: For your final remarks, I'd encourage emphasizing the outcomes in two parts: obs campaings and ML model improvements. Both of which are valuable and complementary, although your ML modelling could be further improved if considering other sources of input data (e.g. reanalysis, NWP, etc.) and additional parameters (e.g. humidity, air density, pressure, temperature, orography and land-use).

A. Done

---

## Author Response (AR1)

Response to reviews

Dear editor, we tried to apply all the referee recommendations. Following are the relevant changes in the manuscript.

1) As advised by the referee, the introduction was reduced to focus on the wind speed modelling. Some contributions were added;
2) A LSTM schematical diagram (Fig. 1) and a table with the LSTM arguments were included (Table A2);
3) Data availability was added (Table A3)
4) Section 2.5 was reduced, as recommended
5) Wind roses include the "%" signal
6) Root Mean Squared Error (RMSE) was added to tables 2, 3 and 4
7) Wind direction temporal series were added to illustrate the diurnal cycle (Fig. A1, A2, A3)
8) Seasonal wind roses were added (Fig. A4)
9) RMSE and MAE were included as subplots of Fig. 6, 7, 8
10) $R^2$ Error bars are shown in Fig. A5, A6 and A7, for better visualization
11) Section 3.2, lines 303 – 314 explain the results from the tests done with samples beyond the validation dataset. Fig. A8 shows the test results from the LSTM. The windshear coefficient ($\alpha$) was computed taking 40 and 60 m wind speed. Afterwards, the PL was estimated. Results also are shown in Fig. A8.

---

## Author Response (AR2)

Response to reviews

Dear editor, we tried to apply all the referee recommendations. Following are the relevant changes in the manuscript.

Concerning report #1:

1) "1%" marker was removed from wind rose plots, as suggested;
2) Bootstrapping was described in section 2.4, line 150. The comments were moved to line 257.

Concerning report #2:

1) Regarding to wind rose plot, was indicated in the text (line 170) that the figure represent the direction where the wind blows from;
2) Edges were rounded as suggested;
3) The repository is indicated in section *Code and data availability*.